# Weight-Space Linear Recurrent Neural Networks

**Roussel Desmond Nzoyem**
University of Bristol
Bristol, UK

**Nawid Keshtmand**
University of Bristol
Bristol, UK

**Enrique Crespo Fernández**
University of Bristol
Bristol, UK

{rd.nzoyemngueguin, enrique.crespofernandez}@bristol.ac.uk

**Idriss Tsayem**
University of Maryland
Baltimore, USA

**Raul Santos-Rodriguez**
University of Bristol
Bristol, UK

**David A.W. Barton**
University of Bristol
Bristol, UK

**Tom Deakin**
University of Bristol
Bristol, UK

## Abstract

We introduce WARP (**W**eight-space **A**daptive **R**ecurrent **P**rediction), a simple yet powerful model that unifies weight-space learning with linear recurrence to redefine sequence modeling. Unlike conventional recurrent neural networks (RNNs) which collapse temporal dynamics into fixed-dimensional hidden states, WARP explicitly parametrizes its hidden state as the weights and biases of a distinct auxiliary neural network, and uses input differences to drive its recurrence. This brain-inspired formulation enables efficient gradient-free adaptation of the auxiliary network at test-time, in-context learning abilities, and seamless integration of domain-specific physical priors. Empirical validation shows that WARP matches or surpasses state-of-the-art baselines on diverse classification tasks, featuring in the top three in 4 out of 6 real-world challenging datasets. Furthermore, extensive experiments across sequential image completion, multivariate time series forecasting, and dynamical system reconstruction demonstrate its expressiveness and generalisation capabilities. Remarkably, a physics-informed variant of our model outperforms the next best model by more than 10x. Ablation studies confirm the architectural necessity of key components, solidifying weight-space linear RNNs as a transformative paradigm for adaptive machine intelligence.

## 1 Introduction

Deep sequence models, which continuously drive progress in machine learning, are limited in their ability to operate outside their training distribution [2; 40; 33]. For instance, subsets of Neural ODE parameters [18] necessitate adaptation via gradient descent to maintain performance on out-of-distribution (OoD) sequences [49; 75]. While effective, their explicit gradient calculation cost has recently catalysed research into **gradient-free test-time adaptation** methods [86; 69; 43]. This surge of interest is embodied by **in-context learning** [60; 86], which has recently been shown to perform test-time adaptation since during inference, it *implicitly* minimises a loss objective using gradient information [94; 102]. Another reason for the poor generalisation of discrete deep sequence models is the inability to inject **domain-specific priors** in their forward pass. In an effort to preserve all desirable traits while unleashing a breadth of possibilities, we combine two of the most powerful emerging deep learning paradigms: weight-space learning and linear recurrence.

**Weight-space learning** — the paradigm that treats the weights and biases of a function approximator as data points for another learning system [85] — offers unprecedented potential for extracting prop-

| State Transition & Decoding in Recurrent Neural Networks | | |
|---|---|---|
| **Standard RNNs** | **Linear RNNs** | **Weight-Space Linear RNNs** |
| $\mathbf{h}_t = f_\Phi(\mathbf{h}_{t-1}, \mathbf{x}_t)$ | $\mathbf{h}_t = A\mathbf{h}_{t-1} + B\mathbf{x}_t$ | $\theta_t = A\theta_{t-1} + B(\mathbf{x}_t - \mathbf{x}_{t-1})$ |
| $\mathbf{y}_t = g_\Psi(\mathbf{h}_t)$ | $\mathbf{y}_t = C\mathbf{h}_{t-1}$ | $\mathbf{y}_t = \mathrm{MLP}_{\theta_t}(\tau)$ |

Figure 1: Background and conceptual comparison between RNN architectures. **Standard** RNNs (e.g. [44; 19]) feature a non-linear transition function $f_\Phi$ unlike their **linear** counterparts (e.g. [37; 77]). Our proposed **weight-space linear** RNNs view their hidden state — denoted as $\theta_t$ — as the parameters of a family of functions. As observed in the bottom-right corner, $\theta_t$ represents, in the general case, the flattened weights of an MLP at time step $t$. Its input $\tau$ is a (concatenation of) coordinate system(s) to maximally make use of the canonical ordering of the sequence.

erties of a trained model solely from its "weights".[1] Applications span from predicting generalisation error [91] and recovering training data [28] to classifying and editing implicit neural representations [24]. With the proliferation of model repositories such as HuggingFace and CivitAI, developing methods that effectively learn directly from weights has become increasingly vital [48]. To date, the literature has predominantly focused on utilizing these weights as inputs and outputs to higher-level models, leaving their potential as *intermediate* representations (e.g., latent vectors, hidden states) in end-to-end training systems unexplored.

Concurrently, **linear** Recurrent Neural Networks (RNNs) have seen a notable resurgence, largely due to their hardware efficiency and the resulting ease of training [22]. Linearity enables advanced sequence parallelisation techniques [87; 71; 100] and has delivered exceptional performance on long-sequence tasks [37; 77]. However, recent findings raise concerns about the information capacity of their compressed state representations [67]. Moreover, a substantial body of work has shown that linear Transformers [50] and State-Space Models (SSMs) [37] — a particular instantiation of linear RNNs — are fundamentally less expressive than the standard non-linear RNNs depicted in Fig. 1 [9; 26; 66]. Taken together, these results strongly suggest that non-linearities are crucial for the expressivity of deep sequence models. They invite the reintroduction of non-linearities into sequence models, while maintaining the hardware-friendly nature of linear RNNs.

The preceding analyses motivate our examination of weight-space linear RNNs. To harness the strengths of its constituting paradigms, we formulate several research questions: ● *Can the weights of an auxiliary function approximator serve as high-resolution hidden states for linear RNNs*? ● *Can that auxiliary function be effectively adapted during inference without requiring gradient computation*? ● *Are the non-linearities in the auxiliary function approximator sufficient to significantly enhance the expressive power of such models*?

We answer these questions in the affirmative by proposing **W**eight-space **A**daptive **R**ecurrent **P**rediction (WARP) models as powerful expressions of weight-space linear RNNs, which we illustrate in Fig. 1. Specifically, our original contributions can be summarised as follows:

(1) We formulate a general framework for sequence modelling in weight-space, blending *linear* recurrence with *non-linear* decoding. Rather than relying on direct inputs, we draw inspiration from the human brain and compute **signal differences** to drive such recurrences. To the best of our knowledge, our framework is the first of its kind to treat weight-space features as intermediate hidden state representations in a recurrence.

(2) To train weight-space linear RNNs, we introduce two parallelisable algorithms: a convolutional mode and an efficient recurrent mode (with and without support for auto-regression) well-suited for noisy sequences. These algorithms unlock three practical use cases: ($i$) **gradient-free adaptation**, i.e., the ability to update critical components responsible for the model's adaptation without requiring gradients; ($ii$) **in-context learning**, i.e., the capacity to recognise input-output patterns in the sequence's context and adapt model behavior without

---

[1]Following the convention from [106], we refer to the learnable parameters of the processed function approximator as 'weights' (or 'weight space' to indicate the space they belong to) and those of the higher-level learning system (e.g., the neural functional) as simply 'parameters'.

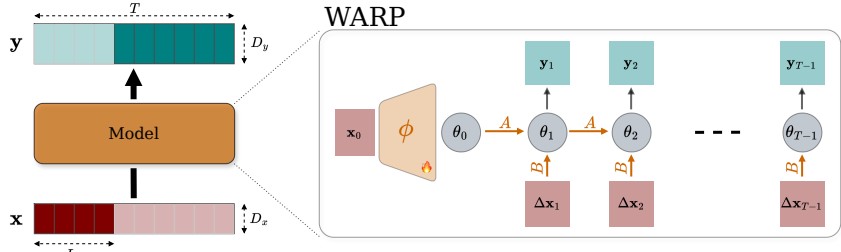

Figure 2: **(Left)** General sequence modelling setting. In the forecasting scenario, for instance, a context of length $L$ informs the prediction of future states. **(Right)** WARP's unfolded recurrence. The initial hypernetwork $\phi$ and transition matrices $(A, B)$ — highlighted in orange — are learnable parameters, fitted via conventional gradient descent.

       finetuning parameters,[2] and $(iii)$ **physics-informed modelling**, i.e., the ability to incorporate domain-specific continuous physical priors in the discrete linear recurrence. This final core application is evidenced in our WARP-Phys model, which achieves an order of magnitude lower error over WARP on a wide set of synthetic dynamical system reconstruction datasets.

(3) We identify an extensive suite of **real-world** benchmarks to evaluate various capabilities of RNNs regarding classification, reconstruction, adaptation, and memory retention. Empirical results demonstrate how WARP consistently matches or outperforms traditional RNNs, SSMs, and Transformer architectures. Remarkably, we push the state-of-the-art by featuring in the top three in 4 out of 6 multivariate time series classification datasets necessitating the understanding of both short- and extremely **long-range** dependencies.

## 2    WEIGHT-SPACE ADAPTIVE RECURRENT PREDICTION (WARP)

This section presents the core ideas underpinning weight-space linear recurrence, our novel framework for deep sequence modelling that operates by directly modulating, in response to sequential input differences, the weights of a function approximator [6]. Out of simplicity and consistency with the related literature in Appendix A, we focus in the remainder of this paper on the WARP model, which modulates a feed-forward neural network [65]. We begin by establishing the problem setting, followed by WARP's architectural and training details.

### 2.1    PROBLEM SETTING

Our framework addresses the general sequence modelling problem, wherein a computational model must establish a mapping from an input $\mathbf{x}_t \in \mathbb{R}^{D_x}$ to a corresponding output $\mathbf{y}_t \in \mathbb{R}^{D_y}$, with $t \in \{0, \ldots, T-1\}$ denoting the time step index. The integer $T > 0$ represents the training sequence length, which remains invariant across all sequences within a training batch.[3] We assume that all input sequences are sampled at the same *uniform* intervals. Ignoring the batch dimension for simplicity, our models establish a mapping from $\mathbb{R}^{T \times D_x}$ to $\mathbb{R}^{T \times D_y}$ (see Fig. 2).

In the regression setting of time series **forecasting**, we have $\mathbf{y}_t \triangleq \mathbf{x}_{t+1}$, as our objective is to predict future tokens conditioned on a preceding sequence of tokens, designated as the "context" $\mathbf{x}_{<L} \triangleq \{\mathbf{x}_t\}_{t \in \{0, \ldots, L-1\}}$, where $L$ denotes the context length. Critically, we desire the ability to perform auto-regressive rollouts during inference. For **classification** tasks, only the final token $\mathbf{y}_{T-1}$ is treated as a softmax-activated logit to assign a label to the sequence.

---

[2]While high-level model parameters must be frozen, in-context learning may still require gradients to finetune some weights.

[3]We note that $T$ may be different for testing sequences. $D_\bullet$ is the dimensionality of the subscripted quantity.

## 2.2 ARCHITECTURE

While traditional recurrent networks update obscure hidden states $\mathbf{h}_t, \forall\, t \in \{1, \ldots, T-1\}$, weight-space linear RNNs such as WARP update the weights and biases of an auxiliary "**root**" neural network $\theta_t$, effectively learning a dynamics model in weight-space (see Figs. 1 and 2). Specifically, we define the recurrence relation and the subsequent decoding:

$$\theta_t = A\theta_{t-1} + B\Delta\mathbf{x}_t, \qquad \text{and} \qquad \mathbf{y}_t = \mathrm{MLP}_{\theta_t}(\tau), \tag{1}$$

where the hidden state $\theta_t \in \mathbb{R}^{D_\theta}$ represents the flattened weights of the root neural network at time step $t$, and $\Delta\mathbf{x}_t = \mathbf{x}_t - \mathbf{x}_{t-1}$ is the input difference. $A \in \mathbb{R}^{D_\theta \times D_\theta}$ is the state transition "**weights-to-weights**" matrix, and $B \in \mathbb{R}^{D_\theta \times D_x}$ the input transition "**data-to-weights**" matrix. To compute the output $\mathbf{y}_t$, the vector $\theta_t$ is unflattened and combined with *non-linear* static activation functions to reconstitute the MLP root network. This decoding function is applied to $\tau$, a **coordinate system** (or a concatenation thereof) that suitably informs the model of the canonical ordering of the sequences at hand. Powerful examples of coordinate systems (see Appendix B.2.1) include normalised pixel locations (for images viewed as sequences), normalised training time $\tau = t/(T-1)$, or the general positional encoding to facilitate generalisation beyond $T$ [92].

Compared to other RNNs, $\theta_t$ plays both the roles of the hidden state and the parameters of the decoder, effectively decoding itself. Such *self-decoding* significantly saves on learnable parameter count.

Importantly, all hidden states can be precomputed efficiently thanks to the *linear* recurrence in Eq. (1), using for instance, the *parallel* "scan" operator [87]. Once materialised, the $\theta_t$ can be reconstituted and self-decoded independently. This allows our model to combine the efficiency of linear recurrence with the expressivity enabled by incorporating non-linearities.

Another key aspect of our formulation is the use of **input differences** $\Delta\mathbf{x}_t$ rather than direct inputs $\mathbf{x}_t$, which is a choice Kidger et al. [54] theoretically motivated for continuous-time RNNs. When inputs change slowly or remain constant, the weight updates become proportionally smaller, and vice-versa. WARP essentially learns to convert input differences into neural network updates, a critical self-supervision ability for continual learning and test-time adaptation [10].

**Architecture of the root network.** The root network $\theta_t$ is implemented as a fixed-width multilayer perceptron (MLP) [65] with a $D_\tau$-dimensional input, and output dimension either $D_y$ or $2 \times D_y$ depending on whether *uncertainty* measures are required in the pipeline. When modelling uncertainty, the network predicts in addition to a mean $\hat{\boldsymbol{\mu}}_t \in \mathbb{R}^{D_y}$, a quantity $\tilde{\boldsymbol{\sigma}}_t \in \mathbb{R}^{D_y}$ on which a positivity-enforcing function is applied to obtain an element-wise standard deviation $\hat{\boldsymbol{\sigma}}_t = \max(\mathrm{softplus}(\tilde{\boldsymbol{\sigma}}_t), \sigma_{\min})$, where $\sigma_{\min}$ is a fixed positive problem-dependent lower bound for numerical stability.

**Initialisation of learnable parameters.** Similar to prior work [58], the state transition matrix $A$ is initialised as the identity operator $I_{D_\theta \times D_\theta}$. This emulates gradient descent and residual connections in ResNets [42], thereby facilitating gradient flow during backpropagation through time [98]. We find that initializing the input transition matrix $B$ as the zero matrix $\mathbf{0}_{D_\theta \times D_x}$ is useful to ensure that the sequence of weights $\theta_t$ does not diverge early on in the training. This strategic initialisation also imposes a critical constraint wherein the initial hidden state $\theta_0$ must encode semantically rich information applicable to the entire sequence.

The initial weights $\theta_0$ are determined by processing the first observation: $\theta_0 = \phi(\mathbf{x}_0)$, where the "initial network" $\phi$ is a hypernetwork [39] defined as a learnable MLP with gradually increasing width (see Fig. 2). On sequence modelling problems with fixed or mildly-varying initial conditions, we sidestep $\phi$ and directly learn $\theta_0$, which is initialised with classical techniques such as Glorot [30] or He [41] (and subsequently flattened into a 1D vector).

## 2.3 TRAINING & INFERENCE

Analogous to SSMs [37] and subsequent linear recurrence architectures [77; 71], WARP supports dual training modes: convolutional and recurrent. The former is accomplished through a systematic unrolling of the linear recurrence formulated in Eq. (1), enabling the derivation of a **convolution** kernel $K$ such that $\theta_{0:T} = K \star \Delta\mathbf{x}_{0:T}$. Comprehensive notations, algorithms, and rigorous mathematical

derivations are elaborated in Appendix B.2.2. In **recurrent** mode, we distinguish the auto-regressive (*AR*) and the relatively memory-expensive[4] *non-AR* settings. The non-AR setting never sees its own predictions, making it ideal for classification tasks wherein $\theta_t(\cdot)$ only generates logits.

The recurrent AR setting is particularly advantageous for noisy forecasting tasks that necessitate accurate modelling of the sequential data distribution $p(\mathbf{y}_t|\mathbf{y}_{<t})$. To mitigate *exposure bias* [84], we implement teacher forcing with scheduled sampling [11], wherein the model incorporates uncertainties by sampling $\hat{\mathbf{y}}_t \sim \mathcal{N}(\hat{\boldsymbol{\mu}}_t, \hat{\boldsymbol{\sigma}}_t^2)$ using the reparametrisation trick [55].[5] Selection between ground truth $\mathbf{y}_t$ and predicted $\hat{\mathbf{y}}_t$ follows a Bernoulli distribution with probability $p_{\text{forcing}}$, which we define as a training hyperparameter. That said, we consistently use $\hat{\mathbf{y}}_{t-1}$ in the input difference seen in Eq. (1).

During inference on regression problems, the model operates fully auto-regressively, i.e., $p_{\text{forcing}} = 1$ within the context window, and $p_{\text{forcing}} = 0$ in the forecast window, regardless of the training mode.

Although other loss functions can be considered, our WARP models are trained by minimizing either the mean-squared error (MSE) for deterministic predictions, or the simplified negative log-likelihood (NLL) for probabilistic predictions:

$$\mathcal{L}_{\text{MSE}} \triangleq \frac{1}{T}\sum_{t=0}^{T-1}\|\mathbf{y}_t - \hat{\mathbf{y}}_t\|_2^2, \qquad \mathcal{L}_{\text{NLL}} \triangleq \frac{1}{T}\sum_{t=0}^{T-1}\left(\frac{\|\mathbf{y}_t - \hat{\mathbf{y}}_t\|_2^2}{2\hat{\boldsymbol{\sigma}}_t^2} + \log\hat{\boldsymbol{\sigma}}_t\right). \tag{2}$$

As for classification problems, we use the categorical cross-entropy $\mathcal{L}_{\text{CCE}} \triangleq \sum_{c=1}^{C}\mathbf{y}^{(c)}\log(\hat{\mathbf{y}}_{T-1}^{(c)})$, where $\mathbf{y}$ is the one-hot encoding of the true label class, and $C$ is the number of classes.

Our learning pseudocodes are detailed in Algorithms 1 and 2 of Appendix B, outlining the strong connection to the *fast weights* and *test-time training* literatures [83; 7; 101]. At each training step, the slow-changing RNN parameters $A, B$ and $\phi$ (or $\theta_0$) are updated *once* using gradient descent to minimise one of the loss objectives above. The fast-changing weights $\theta_t$, however, are updated $T - 1$ times using Eq. (1), i.e., *not* using gradient descent. This distinction is central to our model's gradient-free adaptation process.

## 3 EXPERIMENTS

We evaluate WARP on real-world multivariate time series datasets, 2D images, and physical systems. Our experiments elucidate empirical questions regarding forecasting, classification, and dynamical system reconstruction and generalisation. Additional experiments allow us to demonstrate WARP's in-context learning abilities. Theoretical results are presented in Appendix B.2, and experimental details can be found in Appendix D.

### 3.1 IMAGE COMPLETION, ENERGY PREDICTION & TRAFFIC FORECASTING

In the first part of our experiments, we focus on forecasting applied first to raster-scanned pixel-by-pixel image completion, followed by real-world electricity and traffic flow.

**Image Completion** Image completion is cast as a prediction of pixel intensities. We focus on two datasets: MNIST handwritten digits [59], and the celebrity face attributes CelebA [63]—additional image datasets are considered in Appendix E. 2D images are flattened into 1D sequences with length $T = 784$ for MNIST and $T = 1024$ for CelebA.

Following [81], the completion task is conditioned on contexts of variable length $L$. We compare WARP against long-established baselines

Table 1: Lowest test MSE ($\downarrow$) and BPD ($\downarrow$) achieved on MNIST (Top) and CelebA (Bottom). The best along the columns is reported in **bold**, while the second-best is underlined.

| **MNIST** | $L = 100$ | | $L = 300$ | | $L = 600$ | |
|---|---|---|---|---|---|---|
| | MSE | BPD | MSE | BPD | MSE | BPD |
| GRU | 0.074 | 0.623 | 0.054 | 0.573 | 0.015 | 0.485 |
| LSTM | 0.074 | 0.652 | 0.057 | 0.611 | 0.027 | 0.539 |
| ConvCNP | 0.074 | 0.830 | 0.061 | 0.732 | 0.038 | 0.583 |
| S4 | 0.072 | 0.640 | 0.049 | 0.520 | 0.019 | **0.406** |
| **WARP** | **0.071** | **0.615** | **0.042** | **0.516** | **0.014** | 0.416 |

| **CelebA** | $L = 100$ | | $L = 300$ | | $L = 600$ | |
|---|---|---|---|---|---|---|
| | MSE | BPD | MSE | BPD | MSE | BPD |
| GRU | 0.063 | 24.14 | 0.048 | 60.39 | **0.027** | 71.51 |
| LSTM | 0.064 | 3869 | 0.053 | 7.276 | 0.032 | 7.909 |
| ConvCNP | 0.080 | 1.498 | 0.100 | 39.91 | 0.132 | 248.1 |
| **WARP** | **0.051** | **0.052** | **0.040** | **-0.043** | 0.027 | **-0.162** |

---

[4]Although equal to AR in computational complexity, the recurrent non-AR setting requires more memory because, like the convolutional mode, it materialises all *high-dimensional* hidden states $\theta_t$.

[5]We remark that this sampling is not required during *inference* on smooth sequences like dynamical systems.

like GRU [19] and LSTM [44]; against state-of-the-art (SOTA) SSMs like S4 [37]; and against the ConvCNP convolution-based meta-learning baseline [31] specifically designed for image completion. All models are trained with the NLL loss in recurrent AR mode to ensure fair comparison, and feature nearly the same number of learnable parameters: approximately 1.68M for MNIST, and 2M for CelebA. Results in Table 1 demonstrate the generative performance of WARP as measured by the MSE and the uncertainty-aware bits-per-dimension (BPD) metrics. We focus on the top performing models across three runs, with corresponding qualitative comparisons — best captured by the BPD — in Appendix F. For instance, Fig. 3(a) shows that at small parameter count, WARP is the only model to accurately generate digits without substantial artefacts.

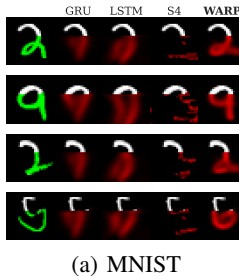
(a) MNIST

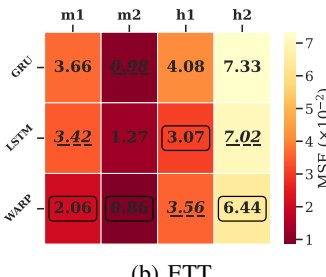
(b) ETT

Figure 3: **(a)** Comparison of a GRU [19], LSTM [44], S4 [81], and WARP on the MNIST image completion task with $L = 300$ initial pixels. All models are roughly at the same size of 1.7M parameters, with architectures described in Appendix C.2. The leftmost column represents target images with context (in white) and ground truths (in green). Predicted forecasts are drawn in red. **(b)** Heatmap of test MSEs (↓) on the ETT task, with best results enclosed and second-best underlined.

**Energy Prediction**     We evaluate WARP's performance on long-range energy forecasting tasks with the Electricity Transformer Temperature (ETT) dataset [108]. Following established methodological protocols [72], we utilise the open-source TSLib[6] to obtain preprocessed data splits which we further normalise using train set statistics (additional data processing details can be found in Appendix C).

The models are tasked with predicting 96 time steps based on a context of length $L = 96$, with performance evaluated using the mean MSE across three runs. The results are shown in Fig. 3(b), where the best along the columns is reported in a box while the second-best is underlined. It demonstrates WARP's superiority, achieving the best performance on all subsets except the ETTh1 subset, where it ranked second. These results are particularly noteworthy given WARP's straightforward design. Indeed, WARP offers an elegant balance between architectural simplicity and predictive power. Additional results on the ETT dataset are presented in Appendix E.

Table 2: Performance on PEMS08 [88]. SOTA baselines leverage spatial information, as reported in [62].

| MODEL | MAE | RMSE |
|---|---|---|
| GMAN [103] | 14.57 | 24.71 |
| D$^2$STGNN [104] | 14.35 | 24.18 |
| STIDGCN [62] | 13.45 | 23.28 |
| **WARP** | **6.59** | **10.10** |

**Traffic Flow Forecasting**     We conduct extensive experiments on the PEMS08 real-world traffic network [88]. The network consists of 170 nodes, from which 3 features are collected at 5-minute intervals over two months. The standard task is to predict the traffic flow for the next hour (12 steps) given the historical data from the previous hour (12 steps). Given its *chunk-wise* forecasting — which significantly differs from the setting in Fig. 2 — we employ the non-AR mode to train and test WARP. Additionally, we preprocess the input sequence with a *non-causal* convolution, as detailed in Appendix D.

As demonstrated in Table 2, our model achieves a MAE of 6.59 and a RMSE of 10.10. These results represent a significant improvement over the current state-of-the-art on the PEMS08 benchmark [62], reducing MAE by over 50% compared to the best-published model. It is particularly noteworthy that our model achieves this performance without using the inherent graph structure, outperforming complex Attention and Graph Neural Network (GNN) architectures that are specifically designed to leverage this spatial information.

---

[6] https://github.com/thuml/Time-Series-Library.git

Table 3: Average test MSE ($\downarrow$) and MAE ($\downarrow$) for dynamical system reconstruction. Best results are reported in **bold**, while the second-best are underlined. All are reported with $\times 10^{-2}$ scaling, except for SINE* with $\times 10^{-4}$. SINE* indicates that metrics are computed upon training on its "Small" data split. WARP-Phys indicates the variant of WARP with physical constraints in the root network.

| | MSD | | MSD-Zero | | LV | | SINE* | |
|---|---|---|---|---|---|---|---|---|
| | MSE | MAE | MSE | MAE | MSE | MAE | MSE | MAE |
| GRU | $1.43 \pm 0.09$ | $5.05$ | $0.55 \pm 0.75$ | $3.27 \pm 0.13$ | $5.83 \pm 0.37$ | $13.1 \pm 0.42$ | $4.90 \pm 0.45$ | $179 \pm 9.23$ |
| LSTM | $1.46 \pm 0.14$ | $5.43 \pm 0.28$ | $0.57 \pm 0.05$ | $3.46 \pm 0.08$ | $6.18 \pm 0.19$ | $13.6 \pm 0.61$ | $9.48 \pm 0.12$ | $248 \pm 3.45$ |
| Transformer | $0.34 \pm 0.12$ | $2.25 \pm 0.42$ | $0.48 \pm 0.24$ | $2.90 \pm 0.32$ | $11.27 \pm 0.62$ | $18.6 \pm 0.49$ | $1728 \pm 10.8$ | $2204 \pm 27.0$ |
| **WARP** | $0.94 \pm 0.09$ | $3.04 \pm 0.11$ | $0.32 \pm 0.02$ | $2.59 \pm 0.07$ | **$4.72 \pm 0.25$** | **$10.9 \pm 0.45$** | $2.77 \pm 0.09$ | $125 \pm 8.46$ |
| **WARP-Phys** | **$0.03 \pm 0.04$** | **$0.66 \pm 0.02$** | **$0.04 \pm 0.01$** | **$0.75 \pm 0.03$** | X | X | **$0.62 \pm 0.01$** | **$6.47 \pm 0.51$** |

## 3.2 DYNAMICAL SYSTEM RECONSTRUCTION

As our final forecasting benchmark, we evaluate WARP's capabilities on dynamical system reconstruction (DSR) [32]. The experiments presented in this section highlight the challenge of OoD generalisation to physical parameters, a research area that has recently experienced a significant surge in interest [76; 15].

We establish four DSR benchmark datasets: ● (1) Mass Spring Damper (MSD) characterises challenging damped oscillatory dynamics through physical parameters $(m, k, c)$, with trajectories of length $T = 256$, of which $L = 100$ states serve as context; ● (2) MSD-Zero is a version of MSD which varies, in addition to the significant relative scales and wide ranges of $(m, k, c)$, the initial condition $\mathbf{x}_0$; ● (3) Lotka-Volterra (LV) is parametrised by coefficients $(\alpha, \beta, \gamma, \delta)$; ● (4) SINE comprises sine curves $\tau \mapsto \sin(2\pi\tau + \varphi)$ with varying phases $\varphi$ (we set $T = 16$ and $L = 1$, resulting in an initial value problem). Each test set incorporates out-of-distribution parameters, except for SINE, which primarily tests model performance under sample size constraints. Comprehensive data generation protocols for all four datasets are detailed in Appendix C. We benchmark against two established RNNs and the Time Series Transformer (TST) from HuggingFace [46] specialised for forecasting. We evaluate WARP in a *black-box* setting — which embeds no explicit physical knowledge in the root network — followed by the more interpretable *grey-box*.

**Black-Box Setting** Our results, presented in Table 3, highlight how weight-space linear RNNs consistently outperform all baseline models across problem domains. Importantly, the standard WARP configuration, which uses a root black-box MLP, ranks within the top two in three out of four problem settings. We observe that TST — denoted simply as Transformer — exhibits significant performance degradation on the SINE* dataset (which comprises only 10 sequences), corroborating the documented limitation of Transformer models to overfit in data-scarce regimes due to their inherently high parameter complexity [27].

**Injecting Physical Bias (Grey-Box)** A principal advantage of WARP is its capacity to incorporate domain-specific knowledge into the root network, exemplified on the SINE* experiment by embedding the explicit mathematical formulation $\tau \mapsto \sin(2\pi\tau + \hat{\varphi})$ in its forward pass, where $\hat{\varphi}$ is predicted by a MLP. The resulting architecture, WARP-Phys, demonstrates substantial performance improvements relative to WARP (more than **one order of magnitude** on MSD). Notably, the incorporation of such a powerful physical prior on SINE* underscores the value of an expressive but data-efficient initial network $\phi$ whose task it is to capture a representation of $\varphi$. Indeed, all models, including WARP and WARP-Phys, perform poorly on the extreme "Tiny" data split (*not* reported in Table 3). We provide additional details as ablations in Appendices E and E.7.

**Repeat-Copy of Physical Systems** We evaluate our model's pattern memorisation capabilities on the Lotka-Volterra (LV) dataset, which constitutes a continuous analogue of the established repeat-copy benchmark [89; 77]. To generate the output shown in red in Fig. 4, we triplicate a concise segment of the input, separating the repetitions by a 10-token long sequence of $-1$s. In this challenging problem, WARP demonstrates superior performance relative to all baselines, with the GRU achieving the second-highest performance metrics (see Table 3). These

Figure 4: Sample LV input/output.

Table 4: Test-set accuracies (↑) averaged over 5 training runs on the UEA classification datasets. Dataset names are abbreviated: EigenWorms (Worms), SelfRegulationSCP1 (SCP1), SelfRegulation-SCP2 (SCP2), EthanolConcentration (Ethanol), Heartbeat, MotorImagery (Motor). Best results are reported in **bold**, and the second-best are underlined.

| | **Worms** | **SCP1** | **SCP2** | **Ethanol** | **Heartbeat** | **Motor** |
|---|---|---|---|---|---|---|
| Seq. length | 17,984 | 896 | 1,152 | 1,751 | 405 | 3,000 |
| # Classes | 5 | 2 | 2 | 4 | 2 | 2 |
| NRDE | $77.2 \pm 7.1$ | $76.7 \pm 5.6$ | $48.1 \pm 11.4$ | $31.4 \pm 4.5$ | $73.9 \pm 2.6$ | $54.0 \pm 7.8$ |
| NCDE | $62.2 \pm 3.3$ | $80.0 \pm 2.0$ | $53.6 \pm 6.2$ | $22.0 \pm 1.0$ | $68.1 \pm 5.8$ | $51.6 \pm 6.7$ |
| LRU | $85.0 \pm 6.2$ | $84.5 \pm 4.6$ | $47.4 \pm 4.0$ | $29.8 \pm 2.8$ | $\underline{78.1 \pm 7.6}$ | $51.9 \pm 8.6$ |
| S5 | $83.9 \pm 4.1$ | $\underline{87.1 \pm 2.1}$ | $55.1 \pm 3.3$ | $25.6 \pm 3.5$ | $73.9 \pm 3.1$ | $53.0 \pm 3.9$ |
| Mamba | $70.9 \pm 15.8$ | $\underline{80.7 \pm 1.4}$ | $48.2 \pm 3.9$ | $27.9 \pm 4.5$ | $76.2 \pm 3.8$ | $47.7 \pm 4.5$ |
| S6 | $85.0 \pm 1.2$ | $82.8 \pm 2.7$ | $49.9 \pm 9.4$ | $26.4 \pm 6.4$ | $76.5 \pm 8.3$ | $51.3 \pm 4.2$ |
| Log-NCDE | $82.8 \pm 2.7$ | $82.1 \pm 1.4$ | $54.0 \pm 2.6$ | $\mathbf{35.9 \pm 6.1}$ | $74.2 \pm 2.0$ | $\underline{57.2 \pm 5.6}$ |
| LinOSS | $\mathbf{95.0 \pm 4.4}$ | $\mathbf{87.8 \pm 2.6}$ | $58.2 \pm 6.9$ | $29.9 \pm 0.6$ | $75.8 \pm 3.7$ | $\mathbf{60.0 \pm 7.5}$ |
| FACTS | $\underline{86.7 \pm 3.0}$ | $73.3 \pm 2.8$ | $\mathbf{70.3 \pm 8.8}$ | $28.2 \pm 3.3$ | $70.3 \pm 8.8$ | $49.8 \pm 3.8$ |
| Griffin | $\underline{79.5 \pm 5.1}$ | $80.0 \pm 1.5$ | $43.1 \pm 5.3$ | $24.0 \pm 3.5$ | $77.7 \pm 2.9$ | $43.8 \pm 3.3$ |
| **WARP** | $70.93 \pm 2.7$ | $83.53 \pm 2.0$ | $\underline{57.89 \pm 1.4}$ | $\mathbf{36.49 \pm 2.8}$ | $\mathbf{80.65 \pm 1.9}$ | $56.14 \pm 5.1$ |

findings suggest that the high-resolution weight-space state representation exhibits enhanced pattern retention capabilities compared to conventional methodologies. We note that this particular evaluation protocol is incompatible with the WARP-Phys variant due to the deliberate introduction of artificial discontinuities in the temporal sequences. Comprehensive analyses of additional results pertaining to this task, alongside other dynamical system reconstruction benchmarks, are presented in Appendix E.

### 3.3 MULTIVARIATE TIME SERIES CLASSIFICATION

We now consider the classification setting. We consider six datasets from the University of East Anglia (UEA) multivariate time series classification archive (UEA-MTSCA) [8]. The six datasets are selected and preprocessed following the criteria of known difficulty for deep sequence models and data abundance, with sequence length ranging from 405 to almost 18k [96]. Our model is compared to both discrete and continuous recurrent baselines [70; 54; 77; 87; 96; 36; 80; 72; 23]. All models are trained, validated, and tested with the 70:15:15 split. Additional details on the dataset preprocessing, the baselines, and the positional encoding used for $\tau$ are provided in Appendix C.

Table 4 presents test accuracy metrics across all six benchmark datasets for WARP (trained in recurrent non-AR mode) and competing methodologies as reported in [96]. Our analysis reveals that WARP demonstrates exceptional performance across the majority of tasks, establishing new state-of-the-art accuracies on the SCP2 Ethanol and Heartbeat datasets, and competitive **top three on four datasets**. Despite not being designed with long-range dependencies in mind, WARP displays impressive potential on extremely long sequences such as EigenWorms and Motor, outperforming established models such as Mamba [36] and NCDE [54], FACTS [72], and Griffin [23]. This overcoming of the well-documented vanishing and exploding gradient problems in recurrent architectures [112] is attributed to our careful initialisation scheme in Section 2.2, and the positional encoding scheme using sines and cosines with variable frequencies [92]. These empirical findings substantiate WARP's efficacy as a robust classification framework for diverse real-world time series applications.

### 3.4 IN-CONTEXT LEARNING WITH RANDOMLY GENERATED KEYS

A key strength of WARP is illustrated in the classical in-context learning (ICL) setting of [102], where the objective is to find a shared vector $\mathbf{w} \in \mathbb{R}^{D_x - 1}$ that linearly maps $N$ randomly generated keys $\mathbf{x}_i \in \mathbb{R}^{D_x - 1}$ to their corresponding values $y_i \in \mathbb{R}^1$. In this setup, WARP learns the weights of the root network that approximate this mapping. We adapt the task by transforming the input sequence into its *cumulative sum* along the temporal dimension, followed by the prediction of the value corresponding to the query $\mathbf{x}_q$ (see Fig. 5(a)). This preserves the underlying function while allowing the model to exploit key-value pairs dependencies. WARP is trained in its recurrent, non-autoregressive mode with a MSE loss over the entire 1D output sequence of length $T = N + 1 = 32$. Critically, we do not

employ a hypernetwork $\phi$ in this task, as $\theta_0$ is fitted directly (see Section 2.2). The results, shown in Figs. 5(b) and 5(c), highlight WARP's ability to perform sub-quadratic in-context learning and generalize effectively.

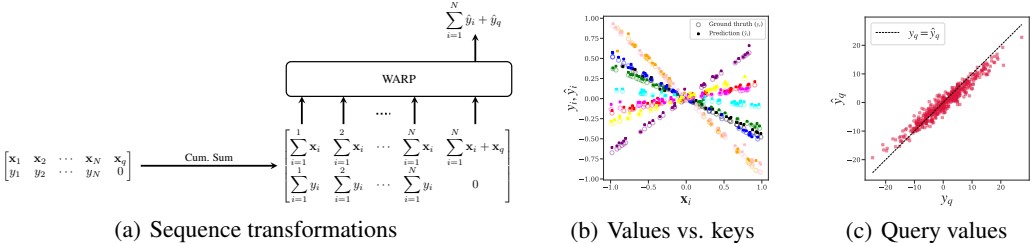

(a) Sequence transformations      (b) Values vs. keys      (c) Query values

Figure 5: Pipeline and results for in-context learning. **(a)** Cumulative sum transformation and subsequent processing of the input matrix. **(b)** Linear mappings learned between scalar keys and values of the same sequences ($D_x = 2$). **(c)** Ground truth vs. query point predictions ($D_x = 8$).

A key advantage of this approach is that once the model has learned from the context, the final root network $\theta_{T-1} : \sum_{i=1}^{N} \mathbf{x}_i + \mathbf{x}_q \mapsto \sum_{i=1}^{N} \hat{y}_i + \hat{y}_q$, which is equivalent to $\theta_{T-1} : \mathbf{x}_q \mapsto \hat{y}_q$, can be extracted. This allows it to process subsequent queries **without** needing to re-evaluate the entire sequence from scratch. This method yields significant computational savings compared to other models capable of ICL [60].

## 4   Discussion & Conclusion

### 4.1   Core Advantages

WARP demonstrates **outstanding results** across a multitude of data modalities, both in-distribution and out-of-distribution, as evidenced by the extensive empirical results on time series forecasting and classification we have presented (see Tables 1 to 4 and Figs. 3(a), 3(b) and 5). Additional results showcasing a 93% classification accuracy on sequential MNIST, along with **ablation studies** and further results on synthetic datasets are provided in Appendix E. Specifically, Appendix E.3 illustrates the excellent **computational efficiency** of our approach, as measured by wall-clock training time per epoch, peak GPU usage, and parameter counts.

By letting the data directly interact with the weights as in Eq. (1), WARP showcases the appealing **in-context learning** ability to fine-tune an auxiliary network **without** gradients at test-time [10; 97]. Additionally, WARP is the latest scientific machine learning [21] technique that seamlessly integrates interpretable **physical knowledge** into its predictions, a feature standard RNNs have overlooked. This demonstratively allows for sample-efficient training and improved generalisation.

Finally, the WARP architecture, through its input difference, bears resemblance to synaptic plasticity in biological neural networks, specifically Spike Timing-Dependent Plasticity [16], wherein the weight of a synaptic connection is strengthened or weakened depending on the time difference between spikes from pre- and post-synaptic neurons. This **neuromorphic quality** enables more biologically plausible learning dynamics.

### 4.2   Limitations

Some design decisions that strengthen WARP equally face limitations that we outline as promising avenues for future work. First, the size of the matrix $A$ limits **scaling to huge root neural networks**. Our experiments conducted on a RTX 4080 GPU with 16GB memory could only support moderate $D_\theta$ values, leaving open the question of how expressive WARP models can become if scaled further. Second, more **theoretical research** is needed to supplement the current state of the weight-space learning literature. Our work remains mostly empirical, despite introducing theory-informed algorithms in Appendix B.2 and leveraging the underpinnings of NCDEs as universal approximators generalizing RNNs in continuous time settings [52]. Lastly, WARP still struggles to achieve SOTA classification performance on **extremely long sequences** with intricate dependencies such as images,

and remains untested on language modalities. Future work would seek first principles to improve long-range performance while reducing the memory footprint of the matrix $A$, by exploring low-rank complex-valued diagonal parametrisations [38], neuron permutation equivariance [105], or block-diagonal decompositions.

### 4.3 CONCLUSION

In this work, we introduced Weight-Space linear RNNs, a novel family of sequence models that operates directly within the weight space of neural networks, offering a distinct paradigm from traditional recurrent architectures. We argue that the high-dimensional weight space can be used for intermediate representations, resulting in "infinite-dimensional" RNN hidden states and high-capacity memory. Our comprehensive experiments demonstrate that our models exhibit superior expressivity and generalisation capabilities, enabling a powerful form of gradient-free adaptation in response to sequential input differences, and showing exceptional abilities when integrating domain-specific knowledge from physical systems. Our framework draws intriguing parallels to neuromorphic learning principles, leading us a step further towards human-level artificial intelligence.

### BROADER IMPACT

While their benefits are evident from Section 4.1, malicious deployment of our self-adaptable models in scenarios they were not designed for could lead to serious adverse outcomes. Additionally, high-energy costs from high-dimensional weight-space computations could increase disparities in our field. To limit the potential for such outcomes and to improve the democratisation of AI, our code is openly available at `https://github.com/ddrous/warp`.

### ACKNOWLEDGMENTS

This work was supported by UK Research and Innovation grant EP/S022937/1: Interactive Artificial Intelligence, EPSRC LEAP Digital Health Hub grant EP/X031349/1, and EPSRC program grant EP/R006768/1: Digital twins for improved dynamic design. We thank Hengshuai Yao and Yasin Abbasi-Yadkori for valuable discussions culminating in ideas that helped improve the appeal and performance of weigh-space linear recurrent neural networks.

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

# Weight-Space Linear Recurrent Neural Networks
## *Supplementary Material*

## A    RELATED WORK

The problem of sequence modelling, long dominated by Transformers [92], is experiencing a renewed focus on recurrent architectures, particularly for their efficiency and unique modelling capabilities [37; 23]. Our model, the Weight-space Adaptive Recurrent Predictor (WARP), intersects with several active research areas.

**Weight-Space Learning**    The concept of leveraging the weight space of neural networks is not new; for instance, optimisers and hypernetworks inherently process weight-space features [3; 107; 39], treating them as *outputs* of a learning algorithm. Other works explore weight features as *inputs* for model analysis [91; 85] or for implicitly representing data [28]. WARP distinguishes itself by directly evolving the root network's weights as its *intermediate* state, without explicitly specifying a test-time loss function to minimise. This test-time regression view is similarly observed with research on linear attention [100; 101; 95] and fast weights programming [7].[7] In the autoregressive forecasting setting, WARP bears striking similarities to WeightFlow [61] which uses graph Controlled Differential Equation [54] to model the continuous-time evolution of the weights, and to the "delta" rule [82], which equally updates weights based on the difference between the prediction and the target. WARP can thus be viewed as a generalisation to broader problem settings that include classification.

**Modern Linear RNNs and SSMs**    Linear RNNs and SSMs have re-emerged as powerful tools, largely due to their parallelisable and hardware-aware training [100; 71], with impressive performance on long sequences. Notable architectures like S4 [37] and Linear Attention [50] have massively catalysed recent advancements. While WARP builds on the efficiency of linear recurrence, its core innovation lies in its unique state parametrisation — rather than solely on the recurrent mechanism — which includes non-linearities for improved expressivity.

**Non-Linear Recurrent Mechanisms**    The landscape of sequence modelling is rich with innovative designs. Hybrid models like Griffin [23] merge recurrences with attention, while Movahedi et al. [71] seek to compute dense linear RNNs from diagonal ones via fixed-point transformations. Frameworks like FACTS introduce structured memories [72]. Brain-inspired architectures [113; 7], including time-varying decoder architectures [45], seek to learn evolving relationships between model inputs and outputs. WARP contributes to this evolving field by introducing a novel mechanism — viewing the RNN hidden state as the weights and biases of a time-varying root neural network — which results in non-linear self-decoding.

**State and Memory in Recurrent Models**    A central debate revolves around the true state-tracking and memory capabilities of various recurrent architectures. While some SSMs and even Transformers face theoretical limitations in solving certain tasks [67; 47], improvements like incorporating negative eigenvalues in linear RNNs aim to enhance state-tracking [34]. Other works explicitly include neural memory modules so that surprising events are more memorable [10]. The growing *test-time training* community [101; 95] proposes to combine recurrence with associative memories for improved sequence modelling. WARP's use of a high-dimensional weight space for its states is a direct attempt to provide richer "infinite-dimensional"[8] memory capacity and more expressive temporal dynamics compared to conventional compressed state representations. This has parallels with the *fast weights* literature [83; 7].

**Gradient-Free Adaptation and Zero-Shot Learning**    Effective adaptation to out-of-distribution dynamics or in continual learning settings is a significant challenge [35]. For instance, standard Neural Ordinary Differential Equations [18] struggle with distribution shifts and need retraining or fine-tuning for adaptation [56; 49]. With its gradient-free formulation, WARP facilitates test-time generalisation — a problem explored in meta-learning frameworks like Neural Context Flows [75] — through differentiable closed-form solvers [12], or in-context learning [94]. WARP can be viewed as a *meta-learning* model given its progressive refinement of a shared initialisation $\theta_0$ at test-time, with strong connections to amortised inference [5].

---

[7]In Fig. 6, we explicitly compare our integration with existing fast weights programming methods.
[8]The hidden state is a *function* which lives in an "infinite-dimensional" space.

**Koopman Operators**  Our method can also be viewed as an application of Koopman operator theory to sequence-to-sequence modelling. As it is the case with nonlinear dynamical systems [57], the challenge is to identify the correct set of infinite-dimensional observable functions (the Koopman eigenfunctions) that linearise the dynamics. WARP addresses this by effectively using the neural network to learn a data-driven approximation of the Koopman operator, a technique explored in modern dynamics and machine learning [64; 68].

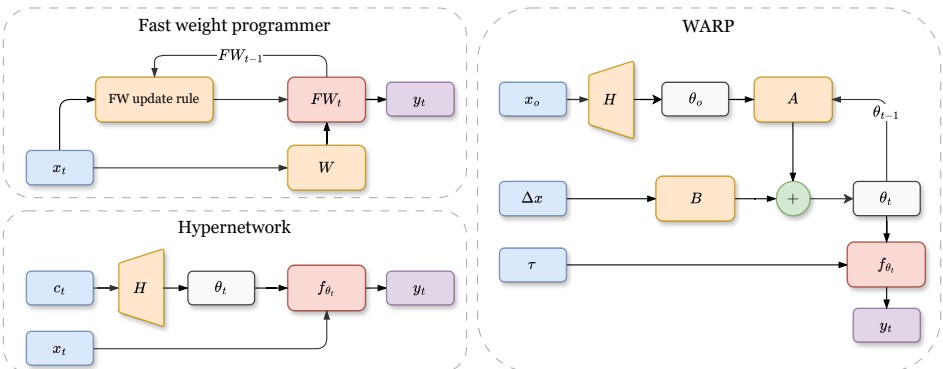

Figure 6: Schematic comparison of adaptive weight architectures. **(Top Left)** The **fast weight programmer** [83] modifies its processing dynamics by iteratively updating the fast weights $FW_t$ using a specialized FW update rule acting on the input $x_t$ and the previous weights $FW_{t-1}$. **(Bottom Left)** A standard **hypernetwork** [17] generates the weights $\theta_t$ for a target function $f_{\theta_t}$ by passing a context code $c_t$ through a higher level network $H$. **(Right)** The proposed **WARP** architecture, where weights are initialized as $\theta_o$ via a hypernetwork $H$ conditioned on $x_o$; subsequent weight updates are driven by a linear recurrence: the previous parameters $\theta_{t-1}$ are processed by block $A$, and input changes $\Delta x$ are processed by block $B$. These components are summed to produce the current weights $\theta_t$, which parameterize the function $f_{\theta_t}$ used to map coordinate inputs $\tau$ to the output $y_t$.

# B    METHODOLOGICAL DETAILS

## B.1    MOTIVATION

The main motivation behind WARP (Weight-space Adaptive Recurrent Prediction) is gradient-free adaptation to out-of-distribution (OoD) settings. Relative to OoD *detection* which has always been a central problem in machine learning spanning decades of research interest [20; 51], OoD *adaptation* is a recent but growing field rich in new and stimulating ideas [4; 75]. WARP mimics the dynamics of an idealised "smooth" gradient descent as observed through a projection of a 4898-dimensional space into a 2-dimensional PCA space in Fig. 7(a). This offers a promising avenue for OoD adaptation with minimal cost.

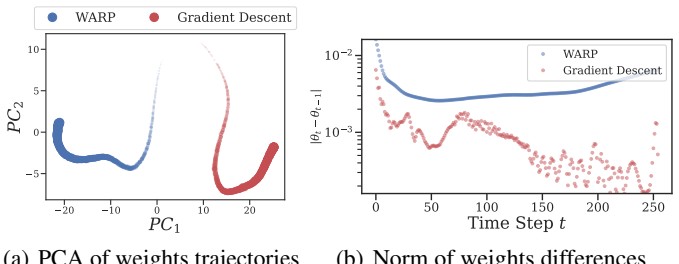

(a) PCA of weights trajectories            (b) Norm of weights differences

Figure 7: **(a)** Principal components of the weight space of WARP vs. weight trajectory fitted via the Gradient Descent strategy on a single trajectory; **(b)** Norm of the difference between updates as we go through the time steps for WARP, and the gradient steps for Gradient Descent.

## B.2    TRAINING ALGORITHMS

---

**Algorithm 1** Recurrent training algorithm for WARP in its non-AR form.

---

**Input:** Training sequences $\{\mathbf{x}_t^i\}_{t\in\{0,...,T-1\}}^{i\in\{0,...,N-1\}}$
**Output:** Trained model parameters $(A, B, \phi)$

**Algorithm:**
1. Initialise $A = I$, $B = \mathbf{0}$, and $\phi$
2. **for** each training epoch **do**
3.     **for** each batch of sequences **do**
4.         **for** each sequence $\{\mathbf{x}_t^i\}_{t\in\{0,...,T-1\}}$ in batch **do**
5.             Initialise $\theta_0^i = \phi(\mathbf{x}_0^i)$
6.             **for** $t = 1$ to $T - 1$ **do**
7.                 Update $\theta_t^i = A\theta_{t-1}^i + B(\mathbf{x}_t^i - \mathbf{x}_{t-1}^i)$
8.                 Compute output $\mathbf{y}_t^i = \text{MLP}_{\theta_t^i}(\tau)$                    (☞ Eqs. (3) to (5))
9.             **end for**
10.            Compute sequence loss $\mathcal{L}^i$ using Eq. (2)
11.        **end for**
12.        Update parameters $A, B, \phi$ using gradient descent
13.    **end for**
14. **end for**

---

### B.2.1    RECURRENT MODE

The recurrent training pipeline is illustrated in its **non-AR setting** in Algorithm 1, where $N$ indicates the total number of instances in the training set, indexed by $i$. The quantity $\tau$ is constructed from the components in Eqs. (3) to (5). Typically, $\tau$ is formed by considering normalised time alone. However, depending on the specific use case, normalised time is concatenated with pixel coordinates (for image data), or with positional encoding using sines and cosines (e.g., for time-series analysis).

- **Normalised Time.** This component consists of the normalised time step, where $T$ is the total sequence length:

$$\tau = \frac{t}{T-1}. \tag{3}$$

- **Normalised Pixel Coordinates.** For image data, spatial information is encoded using normalised pixel coordinates. Given a pixel at position $(w, h)$ in an image of total size $(W, H)$, the coordinates are:

$$\tau = \left[\frac{w}{W-1}, \frac{h}{H-1}\right]. \tag{4}$$

- **Positional Encoding with Sines and Cosines.** The components of this matrix $\tau = PE \in \mathbb{R}^{T \times d}$ are defined as:

$$PE_{(t,k)} = \begin{cases} \sin\left(\frac{t}{C^{2j/d}}\right) & \text{if } k = 2j \\ \cos\left(\frac{t}{C^{2j/d}}\right) & \text{if } k = 2j+1, \end{cases} \tag{5}$$

where $d$ is the encoding dimension, and $C$ is a hyperparameter that controls the frequency of the sinusoidal functions [92].

For the **AR setting** trained with teacher forcing, $\mathbf{x}_t^i$ in line 7 is replaced, with probability $1 - p_{\text{forcing}}$, with a sample from $\mathcal{N}(\hat{\boldsymbol{\mu}}_t, \hat{\boldsymbol{\sigma}}_t^2)$ which is taken element-wise using the classic reparametrisation trick as outlined in Section 2.3.[9] The batch of sequences (lines 4 to 11) is processed in parallel using vectorisation as per the implementation details below.

### B.2.2 Convolutional Mode

Like [37], WARP supports a convolutional training mode where the sequence of weights is computed efficiently using Fast-Fourier Transforms (FFTs) on modern hardware [81] using Theorem 1. We use the Pythonic notation $\mathbf{u}_{0:T} \triangleq \{\mathbf{u}_t\}_{t=0}^{T-1} \in \mathbb{R}^{T \times D_u}$, and the $\star$ to denote the convolution operation. The summarised convolutional training algorithm is provided in Algorithm 2.

**Theorem 1** (Convolution Mode). *Assume $B \in \mathbb{R}^{D_\theta \times D_x}$ is a full row-rank matrix. There exists $\Delta \mathbf{x}_0 \in \mathbb{R}^{D_x}$ and a length-$T$ kernel $K$ such that $\theta_{0:T} = K \star \Delta \mathbf{x}_{0:T}$.*

*Proof.* It follows straightforwardly that the linear recurrence relation $\theta_t = A\theta_{t-1} + B\Delta \mathbf{x}_t$ can be unrolled as

$$\theta_t = A^t \theta_0 + \sum_{\ell=0}^{t-1} A^\ell B \Delta \mathbf{x}_{t-\ell}, \qquad \forall t \in \{1, \ldots, T-1\}. \tag{6}$$

Since $B$ is of full row-rank, the mapping $\mathbf{u} \mapsto B\mathbf{u}$ is surjective, and $\exists \Delta \mathbf{x}_0 \in \mathbb{R}^{D_x}$ such that

$$\theta_0 = B\Delta \mathbf{x}_0. \tag{7}$$

Substituting this into equation 6, we get

$$\theta_t = \sum_{\ell=0}^{t} A^\ell B \Delta \mathbf{x}_{t-\ell}, \qquad \forall t \in \{0, \ldots, T-1\}, \tag{8}$$

from which the large kernel —the sequence of columns of the Kalman controllability matrix [90]— is extracted:

$$K = (B, AB, A^2 B, \ldots, A^{T-1} B), \tag{9}$$

to form the relation

$$\theta_{0:T} = K \star \Delta \mathbf{x}_{0:T} \tag{10}$$

$$\square$$

---

[9]In the main text, the superscripts $i$ were omitted for clarity.

---

**Algorithm 2** Convolutional training algorithm for WARP, where line 6 can be computed with (inverse) FFTs and the convolution theorem. All decoding sequence steps (lines 7-9), as well as the individual sequences (the batch from lines 4-11) are processed in parallel.

---

**Input:** Training sequences $\{\mathbf{x}_t^i\}_{t\in\{0,\ldots,T-1\}}^{i\in\{0,\ldots,N-1\}}$
**Output:** Trained model parameters $(A, B, \phi)$

**Algorithm:**
1. Initialise $A = I$, $B = \mathbf{0}$, and $\phi$
2. **for** each training epoch **do**
3.    **for** each batch of sequences **do**
4.       **for** each sequence $\{\mathbf{x}_t^i\}_{t\in\{0,\ldots,T-1\}}$ in batch **do**
5.          Initialise $\theta_0^i = \phi(\mathbf{x}_0^i)$ and $\Delta\mathbf{x}_0$         (☞ Eq. (7) and Theorem 2)
6.          Compute $\theta_{0:T}^i = K \star \Delta\mathbf{x}_{0:T}^i$         (☞ Eq. (10))
7.          **for** $t = 1$ to $T - 1$ **do**
8.             Compute output $\mathbf{y}_t^i = \text{MLP}_{\theta_t^i}(\tau)$
9.          **end for**
10.          Compute sequence loss $\mathcal{L}^i$ using Eq. (2)
11.       **end for**
12.       Update parameters $A, B, \phi$ using gradient descent
13.    **end for**
14. **end for**

---

In practice, however, we find the assumptions of Theorem 1 too restrictive to be applicable. Indeed, with the weight space typically larger than the input space, i.e. $D_\theta \gg D_x$, the mapping $\mathbf{u} \mapsto B\mathbf{u}$ is not *surjective*. For such cases, we leverage the initial network $\phi$ to enforce additional constraints into the learning process. Theorem 2 guarantees the existence of a suitable initial input difference $\Delta\mathbf{x}_0$ to use as input in the convolution equation 10.

**Theorem 2** (Existence of an Initial Input Difference). *Fix $\phi$ as a locally linear operator with $B = \nabla\phi(\mathbf{x}_0)$, and assume* $\ker \phi \neq \emptyset$. *There exists $v \in \mathbb{R}^{D_x}$ such that $\Delta\mathbf{x}_0 = \mathbf{x}_0 - v$ and Eq. (10) holds.*

*Proof.* The proof is straightforward by remarking that $\theta_0 = \phi(\mathbf{x}_0)$. Using Eq. (7), we find that

$$\theta_0 = B\Delta\mathbf{x}_0$$
$$\Rightarrow \phi(\mathbf{x}_0) = 0 + B\Delta\mathbf{x}_0$$

Since $\ker \phi \neq \emptyset$, $\exists v$ such that $\phi(v) = 0$, and since we've fixed $B = \nabla\phi(\mathbf{x}_0)$, this leads to

$$\phi(\mathbf{x}_0) = \phi(v) + \nabla\phi(\mathbf{x}_0)\Delta\mathbf{x}_0.$$

Since $\phi$ is locally linear, this relation can be identified with its unique first-order Taylor expansion near $\mathbf{x}_0$, from which we identify $\mathbf{x}_0 = v + \Delta\mathbf{x}_0$; or equivalently $\Delta\mathbf{x}_0 = \mathbf{x}_0 - v$.

$\square$

### B.3 IMPLEMENTATION CAVEATS

The difference between a successful WARP training and a failure may lie in small implementation details. We recommend clipping several quantities to increase the chances of success.

**Prediction Clipping** During our training, we found it important to constrain the outputs of the root network to avoid divergence and blow-up. This can be achieved through a final activation applied to the mean component of the output, with e.g. *min-max* symmetric clipping:

$$\mathbf{x}_t \mapsto \max(\min(\mathbf{x}_t, d_{\text{lim}}), -d_{\text{lim}}),$$

with hyperparameter $d_{\text{lim}} > 0$. Another powerful approach which has demonstrated great success in the realm of Transformers is the *dynamic tanh* [109] with learnable scalars $a, b, \alpha, \beta$:

$$\mathbf{x}_t \mapsto \alpha \tanh \left( \frac{\mathbf{x}_t - b}{a} \right) + \beta,$$

with $(a, \alpha)$ initialised as the largest value encountered in the training datasets, and $(b, \beta)$ both as zero. This ensures output scaling that is consistent in shape with the classical $\tanh$ activation.

**Weight Clipping** In some problems like MNIST, we found it not enough to constrain the root's output within a certain bound, as the predictions kept diverging. In such cases, mechanisms like directly clipping the weights in between time steps provided an additional form of non-linearity helpful for the model. Our weight clipping strategy differs from traditional approaches discussed in continual learning contexts [29] as it does not consider initialisation:

$$\theta_t = \texttt{clip}(\theta_t, -w_{\text{lim}}, w_{\text{lim}}),$$

where $w_{\text{lim}}$ is a hyperparameter, and $\texttt{clip}$ is a shorthand for *min-max* clipping as discussed above. This clipping operation serves as an implicit activation function in weight space, preventing unbounded growth in weight values and stabilizing training.

**Gradient Clipping** As customary with recurrent networks training with the backpropagation through time algorithm [98], we observed the classical problem of exploding gradients [112], which was mitigated by clipping the gradient norms within a specific bound captured by $g_{\text{lim}} = 10^{-7}$.

# C  DATASETS, BASELINES & METRICS

## C.1  DATASETS

Table 5: Problems and their corresponding training datasets with specifications. Details about the UEA datasets are presented in Table 4 and not repeated here. The term *(varies)* indicates that further splits of the datasets were made.

| PROBLEM | DATASET | # SAMPLES $N$ | SEQ. LENGTH $T$ | CONTEXT LENGTH $L$ | # FEATURES $D_x$ |
|---|---|---|---|---|---|
| 2D Images | MNIST | 60,000 | 784 | *(varies)* | 1 |
| | Fashion MNIST | 60,000 | 784 | *(varies)* | 1 |
| | CelebA | 162,770 | 1,024 | *(varies)* | 3 |
| ETT | m1 | 34,369 | 192 | 96 | 7 |
| | m2 | 34,369 | 192 | 96 | 7 |
| | h1 | 8,449 | 192 | 96 | 7 |
| | h2 | 8,449 | 192 | 96 | 7 |
| Dynamical Systems | MSD | 20,480 | 256 | 100 | 2 |
| | MSD-Zero | 20,480 | 256 | 100 | 2 |
| | LV | 15,000 | 256 | 100 | 2 |
| | SINE | *(varies)* | 16 | 1 | 1 |
| | Spirals | 10,000 | 64 | 64 | 2 |

We describe various datasets used in this paper. Our description delves into the details of pre-existing datasets and the data generation script of synthetic toy datasets. This section complements the summary we provided in Table 5.

**Image Datasets**  Both MNIST [59] and Fashion MNIST [99] datasets were loaded using the well-known PyTorch interface [78]. The values were then normalised so that pixel values ranged $[-1, 1]$. The CelebA dataset [63] was loaded using the API from [74] itself inspired by [111]. Training was performed on the train sets (see attributes in Table 5), with validation and testing on the predefined test sets.

**Electricity Transformer Temperature (ETT)**  For the electricity data [108], we further normalised the preprocessed data from TSLib to place all values in the range $[-1, 1]$ in order to facilitate learning dynamics. We did not use the predefined "test" set because of its 144-step-long forecast window, which is much longer than the 96 steps all models saw during training. Consequently, we used the "validation" set to evaluate our models as well as all the baselines.

**University of East Anglia (UEA)**  On the UEA dataset [8], we follow the procedure from [96] and reuse the same dataset. (We note that this exact experimental protocol was recently observed in [80]). We extracted the necessary scripts for reproducibility and provide them as part of our code under appropriate license.

**Dynamical Systems**  Continuous autonomous dynamical systems can be conceptualised as multivariate time series governed by a deterministic vector field $(\mathbf{x}_\tau, p) \mapsto \dot{\mathbf{x}}_\tau$, with $p$ encompassing physical parameters affecting the dynamics. Given an initial condition $\mathbf{x}_0$, one can systematically simulate and subsample the trajectory $\mathbf{x}_{0:T}$. To complement our description in Section 3.2, we provide the vector field used for each dataset in Table 6. We summarise their physical parameter ranges in Table 7. All trajectories are obtained with SciPy's 'RK45' adaptive time-stepping numerical integrator [93]. The five training data splits of the SINE are "Tiny", "Small", "Medium", "Large", "Huge", with respectively 1, 10, 100, 1k, and 10k samples. All datasets are normalised and placed within $[-1, 1]$, which is calculated using the train set statistics. Comprehensive data generation scripts with physical parameters for all four datasets are provided in our code.

**Spirals**  The Spirals dataset is an additional dynamical system dataset for binary classification tasks. The training data consists of 10,000 samples, where each sample is a spiral trajectory represented as a sequence of 64 2D points ($x, y$ coordinates). Half of the dataset contains clockwise spirals (labelled as 0), while the other half contains counterclockwise spirals (labelled as 1). The spirals are generated using sine and cosine functions with random phase offsets, and the amplitude decreases with time to

Table 6: List of considered dynamical systems with their vector fields and/or flow maps.

| DATASET | PHYSICAL PARAMETERS | VECTOR FIELD | FLOW MAP |
|---|---|---|---|
| Mass-Spring-Damper (MSD) | $m, k, c$ | $\begin{cases} \dot{x}_1 = x_2 \\ \dot{x}_2 = -\dfrac{k}{m}x_1 - \dfrac{c}{m}x_2 \end{cases}$ | Not used |
| Lotka-Volterra (LV) | $\alpha, \beta, \gamma, \delta$ | $\begin{cases} \dot{x}_1 = \alpha x_1 - \beta x_1 x_2 \\ \dot{x}_2 = \delta x_1 x_2 - \gamma x_2 \end{cases}$ | Non-closed form solution |
| Sine Curves (SINE) | $\varphi$ | Not used | $x_1(t) = \sin(2\pi t + \varphi)$ |

Table 7: Parameter ranges of several dynamical systems for Train and Test datasets. Test set parameter ranges induce OoD trajectories, except for the SINE cases. The relative scale and broad range of parameters values for the MSD problem make this task extremely challenging.

| DATASET | TRAIN PARAMETER RANGES | TEST PARAMETER RANGES |
|---|---|---|
| Mass-Spring-Damper (MSD) | $\begin{cases} m \in [0.02, 0.04] \\ k \in [4, 16] \\ c \in [0.01, 0.2] \end{cases}$ | $\begin{cases} m \in [0.01, 0.05] \\ k \in [2, 18] \\ c \in [0.01, 0.3] \end{cases}$ |
| Lotka-Volterra (LV) | $\begin{cases} \alpha \in [20, 50] \\ \beta \in [80, 120] \\ \gamma \in [80, 120] \\ \delta \in [20, 50] \end{cases}$ | $\begin{cases} \alpha \in [10, 60] \\ \beta \in [70, 130] \\ \gamma \in [70, 130] \\ \delta \in [10, 60] \end{cases}$ |
| Sine Curves (SINE) | $\varphi \in [-\pi/6, \pi/6]$ | $\varphi \in [-\pi/6, \pi/6]$ |

create the spiral effect. This dataset serves as a powerful test case for dynamics classification; it was inspired by [53],[10] with a few samples visualised in Fig. 8.

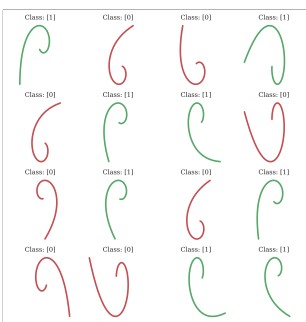

Figure 8: Visualisation of a few samples of the Spirals datasets. For each sample, we plot the $y$ and the $x$ sequences of coordinates against each other to observe the (counter-)clockwise direction.

## C.2 BASELINES

All models are trained based on the same hyperparameter tuning protocol in order to ensure fair comparability.

**Standard RNNs**  We consider two powerful RNN baselines including the Gated Recurrent Unit (GRU) [19] and the Long Short-Term Memory (LSTM) [44]. Both are trained in recurrent AR mode for forecasting problems and recurrent non-AR mode for classification. Both are unidirectional and have a single layer to match our WARP model. Depending on the experiment, we vary their hidden size to match the total parameter count of WARP. The remainder of the experiment details such as

---

[10]An example usage can be found at `https://docs.kidger.site/diffrax/examples/neural_cde/`

training procedure are presented in Appendix D. We attach both implementations, using Equinox [53], to our code.

**Time Series Transformer (TST)**    We consider the Time Series Transformer from HuggingFace [46]. This baseline provides a SOTA baseline, leveraging one of the most transformative sequence mixing processes to date: Attention. The specific model used was the *TimeSeriesTransformerForPrediction*, which adds a distribution head on top of the vanilla encoder-decoder Transformer [92]. This means that the model learns a distribution from which we take the mean to be used for time-series forecasting. The next token prediction is obtained by randomly sampling a window of context length $L$ plus prediction length $T - L$ from the target time series. This prediction window is subsequently masked for the next token prediction task.

**Convolution Conditional Neural Process (ConvCNP)**    The ConvCNP [31] is an encoder-based meta-learning approach that doesn't require gradients in order to adapt to novel scenarios. The ConvCNP is trained for on-the-grid image completion with 100 random shots (time steps. We adapt the data loading process to allow the ConvCNP to operate on raster-scan-ordered pixels at test time.

**Structured SSM (S4)**    We use the powerful Structured State Space Model S4 with the implementation of [81]. We particularly apply it to the MNIST experiment, where the goal is to forecast a contiguous range of future predictions given a range of past contexts. To that end, we simply concatenate the entire context with a sequence of masks set to the length of the forecast window. This input is a single sequence of length $T$ that is run through the deep S4 model, which maps to an output of length $T$. We then use the last $T - L$ tokens as the forecasted predictions. Unlike other models in this work, the MNIST image completion problem with S4 is trained with a 256-way cross-entropy loss, as pixel intensities take integer values in the range $[0, 255]$. This limits the fair applicability of S4 on the CelebA dataset, since its images contain all three RGB channels.

**Neural Controlled Differential Equation (NCDE)**    NCDEs or (Neural CDEs) [54] provide a continuous-time framework for processing irregularly-sampled time series by interpreting the data as a continuous path. By using the path as a control for a neural differential equation, NCDEs can naturally handle missing data and irregular sampling. The continuous nature of NCDEs makes them a strong baseline exclusively for classification tasks.

**Baselines for classification**    Mamba, S6, Log-CDE, NRDE, NCDE, LRU were all reported from [96], where we direct the reader for further details. We reused the results and the conclusion from that work, as was done by Rusch & Rus [80]. The LinOSS baseline [80] reported in Table 4 corresponspods to the more powerful LinOSS-IM variant. We used the official implementations of FACTS [72] and Griffin [23]. Griffin's model size was reduced to 5k to fit within our compute budget.

## C.3 METRICS

**Bits Per Dimension (BPD)**    The (BPD) is used to evaluate the quality of generative models, particularly for images. It quantifies how many bits are needed on average to encode each dimension (e.g., pixel) of the data, with lower BPD values indicating a better model. The BPD is derived from the negative log-likelihood (NLL) of the data under the model's predicted distribution. For a given ground truth pixel value $\mathbf{y}_t$ and its corresponding predicted mean $\hat{\mathbf{y}}_t$ and standard deviation $\hat{\boldsymbol{\sigma}}_t$, the overall NLL over the image is calculated as:

$$\text{NLL} \triangleq \frac{1}{T} \sum_{t=0}^{T-1} \frac{1}{2} \log(2\pi\hat{\boldsymbol{\sigma}}_t^2) + \frac{1}{2} \frac{(\mathbf{y}_t - \hat{\mathbf{y}}_t)^2}{\hat{\boldsymbol{\sigma}}_t^2}.$$

The BPD is obtained by converting the NLL from natural units of information to bits:

$$\text{BPD} = \text{NLL} \times \log_2(e).$$

**Mean Absolute Error (MAE)**    The MAE measures the average magnitude of the errors in a set of predictions. For a sequence of true values $\mathbf{y}_t$ and predicted values $\hat{\mathbf{y}}_t$, the MAE is given by:

$$\text{MAE} \triangleq \frac{1}{T} \sum_{t=0}^{T-1} |\mathbf{y}_t - \hat{\mathbf{y}}_t|.$$

**Mean Absolute Percentage Error (MAPE)**    The Mean Absolute Percentage Error (MAPE) expresses the average absolute percent error. The MAPE is given in percentage points by:

$$\text{MAPE} \triangleq \frac{100}{T} \sum_{t=0}^{T-1} \left| \frac{\mathbf{y}_t - \hat{\mathbf{y}}_t}{\mathbf{y}_t} \right|.$$

# D    EXPERIMENTAL DETAILS

We begin this section by sharing experimental details shared across all problems. Subsequent subsections will delve into the specifics of each completion, forecasting or classification problem.

**WARP setup.**    Although specific details may vary depending on the problem, the root network is consistently chosen as an MLP for all problem sets in this paper. Given the quadratic memory cost $O(D_\theta^2)$, we can vary its layers to balance batch size with capacity. Image completion and forecasting problems use the ReLU activation [1], while smooth dynamical system reconstruction uses the Swish [79]. Complete details on the root network are given in Table 8. The initial hypernetwork $\phi$ —used for all problems except Image Completion, MSD, and LV— is made up of two hidden layers of width $h_{\text{in}}/3 + 2h_{\text{out}}/3$ and $2h_{\text{in}}/3 + h_{\text{out}}/3$ neurons, respectively. The positive integers $h_{\text{in}}$ and $h_{\text{out}} = D_\theta$ are the number of input and output neurons, respectively.

Table 8: Root MLP configurations for the datasets in each problem.

| PROBLEM | DATASET | WIDTH | DEPTH | ACT. FUNCTION |
|---|---|---|---|---|
| 2D Images | MNIST | 24 | 3 | ReLU |
| | Fashion MNIST | 24 | 3 | ReLU |
| | CelebA | 24 | 3 | ReLU |
| ETT | m1 | 148 | 1 | ReLU |
| | m2 | 148 | 1 | ReLU |
| | h1 | 148 | 1 | ReLU |
| | h2 | 148 | 1 | ReLU |
| Dynamical Systems | MSD | 48 | 3 | Swish |
| | MSD-Zero | 48 | 3 | Swish |
| | LV | 48 | 3 | Swish |
| | SINE | 48 | 3 | Swish |
| | Spirals | 24 | 1 | Swish |
| UEA | Worms | 128 | 1 | ReLU |
| | SCP1 | 48 | 2 | ReLU |
| | SCP2 | 48 | 2 | ReLU |
| | Ethanol | 32 | 2 | ReLU |
| | Heartbeat | 72 | 2 | ReLU |
| | Motor | 32 | 2 | ReLU |

**WARP-Phys setup.**    For the SINE experiment, the root network predicts the phase $\hat{\varphi}$ to feed into the sinusoid $\tau \mapsto \sin(2\pi\tau + \hat{\varphi})$. For the challenging MSD and MSD-Zero problems, we embed knowledge of the general analytical solution and the initial condition with $\tau \mapsto E(\tau)\mathbf{x}_0$, where $E(\cdot) \in \mathbb{R}^{2\times2}$ with its four coefficients predicted by the root network, and $\mathbf{x}_0$ is known throughout. $E(\tau)$ is viewed as the exponential of $\tau A$, where $A$ is the constant matrix characterizing the mass-spring-damper dynamics: $A = \left(\begin{smallmatrix} 0 & 1 \\ -k/m & -c/m \end{smallmatrix}\right)$ [73]; its poor conditioning — a consequence of the large parameter scales and ranges detailed in Table 7 — would destabilise the training if $A$ was directly learned. We note that stronger levels of physics may be embedded into the root network: predicting rescaled time-invariant constants $(m, k, c)$, parameterising the signal as damped sinusoids, eigen-decomposition, etc. The physics-informed strategy we present is the one that produced the biggest improvement over WARP in our experiments.

**Optimisation & Core baselines.**    Our WARP framework (along with our custom GRU, LSTM, ConvCNP, and Neural CDE) is implemented with the JAX framework [14] and its ecosystem: Equinox for neural network definitions [53], and Optax for optimisation [25]. We use the AdaBelief optimiser [110], and we clip the gradient norm with $g_{\text{lim}} = 10^{-7}$. We apply the "reduce on plateau" rule where the learning rate is divided by 2 if the average loss[11] doesn't evolve after 20 epochs. For most problems, we set the initial learning rate at $10^{-5}$. All GRU and LSTM models have a single layer to match WARP. We tweak their hidden size rather than the number of layers in order to increase or reduce parameter count, thus keeping in check the complexity of the models under consideration (see Table 9).

---

[11]The average being calculated over 50 iterations.

Table 9: Size of the hidden state in standard RNNs for each dataset.

| PROBLEM | DATASET | LSTM HIDDEN UNITS | GRU HIDDEN UNITS |
|---|---|---|---|
| 2D Images | MNIST | 750 | 750 |
| | Fashion MNIST | 650 | 750 |
| | CelebA | 700 | 825 |
| ETT | m1 | 920 | 920 |
| | m2 | 920 | 920 |
| | h1 | 920 | 920 |
| | h2 | 920 | 920 |
| Dynamical Systems | MSD | 2450 | 2850 |
| | MSD-Zero | 2450 | 2850 |
| | LV | 2450 | 2850 |
| | SINE | 2280 | 2280 |

**TST and S4 baselines setup.** As for the TST model for forecasting, it is implemented in PyTorch [78] and uses the AdamW optimiser with a constant learning rate of $6 \times 10^{-4}$, beta values of $0.9$ and $0.95$ and weight decay coefficient of $10^{-1}$. For the S4 model, we used 6 layers, each with a hidden state of size 512, a batch size of 50, a learning rate of $10^{-3}$, and the traditional weight decay with coefficient 0.05. Like [81], we used prenormalisation, but we did not use dropout. These hyperparameters were selected based on the validation set when available, and the test set if not (e.g., all synthetic toy problems).

**Hardware.** The WARP, GRU, LSTM, ConvCNP and S4 models are run on a workstation fitted with a RTX 4080 GPU with a memory capacity of 16 GB. The TST was trained on a RTX 3090 GPU with 24 GB memory.

## D.1 IMAGE COMPLETION

On these problems, since trained with the NLL loss from Eq. (2), we use a dynamic tanh activation function on the mean prediction, with $(a, b, \alpha, \beta)$ initialised as $(1, 0, 1, 0)$. Only experiments run on MNIST and Fashion MNIST use weight clipping, while CelebA does not.[12] For CelebA, we use $\sigma_{\text{lim}} = 10^{-4}$ (see Section 2.2) while we find the unusually large $\sigma_{\text{lim}} = 0.5$ suitable for MNIST and Fashion MNIST. We compare WARP to the MNIST baselines roughly at the same parameter counts: GRU (1.694 M), LSTM (1.696 M), S4 (1.717 M), and WARP (1.687 M). We train for 200 and 250 epochs in batches of 640 and 1256 for (Fashion) MNIST and CelebA, respectively. We apply the recurrent AR mode with $p_{\text{forcing}} = 0.15$, while directly feeding the mean prediction back into the recurrence (i.e., the reparametrisation trick is disabled both during training and inference).

As inputs to the root network, while the normalised pixel coordinates are better suited for this task, we report our state-of-the-art results using the normalised time $\tau = 1/(T - 1)$. In fact, all results presented in the paper only use the normalised time coordinate system, except for the time series classification on the UEA dataset presented below.

## D.2 IMAGE CLASSIFICATION

For this task, we use the same hyperparameters as the MNIST task described above, with the only difference that the training is now performed in recurrent non-AR mode.

## D.3 ENERGY FORECASTING

For these experiments, all models share identical architectures across the four datasets (ETT-h1, ETT-h2, ETT-m1, ETT-m2). The learning rate differs between hourly and minute-level datasets: $10^{-5}$ for h1/h2 and $10^{-4}$ for m1/m2. For hourly datasets, we train for 500 epochs, while minute-level datasets require only 250 epochs (corresponding to roughly 1.5 hours of training). All models are trained with batch size 3600 in autoregressive mode with stochastic sampling (non-AR), and

---

[12]Apart from (Fashion) MNIST, no other problem in this work used weight clipping.

$p_{\text{forcing}} = 0.25$. No final activation is applied to the root network's mean output, while the typical positivity-enforcing from Section 2.2 is applied to the standard deviation with $\sigma_{\text{lim}} = 10^{-4}$.

### D.4 TRAFFIC FLOW FORECASTING

Our model architecture disregards the explicit spatial connectivity provided in the PEMS08 dataset [88]. Instead, we consider the features from all nodes independently, creating a flattened feature vector for each time step. This results in an input of shape (12, 510), where 12 is the number of historical time steps and 510 represents the 170 nodes, each with 3 features. Before the input sequence is used in the linear recurrence, its features are transformed by a 1D-convolution with 510 input channels, 4080 output channels, and a kernel length of 36. The model is trained to predict a single feature per node for the future 12 time steps.

### D.5 IN-CONTEXT LEARNING

The setting is the elegant in-context learning setting developed by [102], where the goal is learn the linear mapping between several key-value pairs. The keys $\{\mathbf{x}_i\}_{i=1,\ldots,N}$ are vectors of dimension $D_x - 1$, and the values $\{y_i\}_{i=1,\ldots,N}$ are scalar, both concatenated to form a state of dimension $D_x$. A final query key is given, and the model must predict its corresponding value. This missing value is substituted by 0 in the input sequence, as depicted in Fig. 5(a).

Importantly, to retain consistency across the literature, we preserve the notations from [102], even though they conflict with those established in our problem setting in Section 2.1. To revert back to our original setting, one can replace the existing inputs with $\mathbf{x}_t \triangleq \text{concat}(\mathbf{x}_{t+1}, y_{t+1})$, for $t = 0, \ldots, T-2$; and $\mathbf{x}_{T-1} \triangleq \text{concat}(\mathbf{x}_q, 0)$. As for the outputs, $\mathbf{y}_t \triangleq y_{t+1}$, for $t = 0, \ldots, T-2$; and $\mathbf{y}_{T-1} \triangleq y_q$.

### D.6 DYNAMICAL SYSTEM RECONSTRUCTION

For the dynamical system reconstruction tasks, since uncertainties are not required, models are trained without NLL loss (i.e., with the MSE loss defined in Eq. (2)). Consistent across all dynamical system experiments, no weight clipping is employed, and the predictions are enforced in the range $[-1, 1]$ with a unit-initialised dynamic tanh. All losses are computed on the normalised test set.

**Mass-Spring-Damper (MSD and MSD-Zero)** For both the MSD and its MSD-Zero variant, the experimental setup is largely identical. A learning rate of $10^{-5}$ is used. Training proceeds for 1000 epochs using a batch size of 1024. WARP, GRU, and LSTM models are trained in an auto-regressive mode with a teacher forcing probability $p_{\text{forcing}} = 0.25$.

**Lotka-Volterra (LV)** The LV experiment is performed for 1500 epochs with a batch size of 1024. Training is conducted with a teacher forcing probability $p_{\text{forcing}} = 1.0$, meaning the model is always fed the ground truth inputs during training. This is because this is a memorisation task, and the goal is for the model to predict the next token *knowing* the previous one. LSTM and GRU use the same hyperparameters, except with hidden states of sizes 2450 and 2850 respectively (see Table 9).

**Sine Curves (SINE)** Across the various SINE datasets (Tiny, Small, Medium, Large, Huge), a consistent configuration is maintained. The learning rate is set to $10^{-5}$. Models are trained for 1000 epochs in a single batch (as large as 10000 on Huge). Similar to MSD, training is autoregressive with $p_{\text{forcing}} = 0.25$. No final activation is applied to the root network's mean output. The inference process for SINE datasets begins with a very short context, of just 1 time step.

### D.7 TIME SERIES CLASSIFICATION

For time series classification tasks, encompassing both the UEA datasets and the Spirals dataset, models are consistently trained in the non-AR mode, with the categorical cross-entropy loss. Across all these classification experiments, root weight are evolved without weight clipping, and no dynamic tanh activation is applied to their final outputs. Key training hyperparameters exhibit some variation across these diverse datasets: the learning rate is $10^{-5}$ for the Spirals dataset and most UEA datasets

(e.g., Ethanol, Heartbeat, Motor, SCP1, SCP2), with the Worms dataset being an exception at $10^{-6}$. The number of training epochs varies widely, ranging from 800 for the Worms dataset, 4000 for Spirals, up to 6500 for the Ethanol UEA dataset, with other UEA datasets generally trained for several thousand epochs. Given our limitation of 16GB available VRAM memory, batch sizes also differ significantly; for instance, the Worms dataset uses a batch size of 40, other UEA datasets use batch sizes typically in the hundreds (from approximately 280 to 560), and the Spirals dataset employs a large batch size of 10000. Regarding data preprocessing, input data normalisation is applied for several UEA datasets (specifically Ethanol, Heartbeat, SCP1, and SCP2), but it is not used for others like EigenWorms and MotorImagery, nor is it required for the Spirals dataset.

This task uses positional encoding in addition to normalised time. The dimension $d$ and the denominator constant $C$ of the positional encoding defined in Eq. (5) [92] and used in concatenation with the normalised time on the UEA dataset, are presented in Table 10.

Table 10: Hyperparameters for positional encoding on the UEA datasets.

|  | Worms | SCP1 | SCP2 | Ethanol | Heartbeat | Motor |
|---|---|---|---|---|---|---|
| Dimension $d$ | 20 | 10 | 10 | 10 | 10 | 10 |
| Denominator constant $C$ | 20 | 10 | 10 | 10 | 5 | 10 |

# E  ADDITIONAL RESULTS

## E.1  2D IMAGE EXPERIMENTS

Similar to MNIST image completion, we train WARP, LSTM and GRU to generate items of clothing (Fashion MNIST). The results, presented in Table 11 confirm the potency of our framework, as previously evoked in Section 3.1. We perform an additional classification on the sequential MNIST dataset, where we observe a 99.93% accuracy on the subsampled grayscale images in $\mathbb{R}^{14 \times 14 \times 1}$.

Table 11: Best test-set MSEs and BPDs on Fashion MNIST across 3 runs with different seeds.

| METHOD | MSE | BPD |
|--------|-----|-----|
| GRU | 0.078 | 0.66 |
| LSTM | 0.082 | 0.73 |
| **WARP** | 0.064 | 0.59 |

Table 12: Best accuracies and walltime comparison for Spirals classification across 3 runs.

| METHOD | ACCURACY (%) | WALL TIME / EPOCH (SECS) |
|--------|--------------|--------------------------|
| Neural CDE | 100.0 | 0.12 |
| **WARP** | 99.96 | 0.41 |

## E.2  SPIRALS & NEURAL CDES

Table 12 reveals several limitations of WARP on the toy Spirals dataset originally introduced to test Neural CDEs [52]. We find that at the same parameter count, WARP not only struggles to achieve 100% accuracy, but is also roughly $4\times$ slower, despite being implemented in the same conditions as the Neural CDE.

## E.3  COMPUTATIONAL EFFICIENCY COMPARISON

To provide a comprehensive analysis of computational efficiency, we evaluate several key performance metrics for WARP and our baseline models. The experiments were conducted on an NVIDIA RTX 4080 GPU, ensuring a consistent hardware environment for all comparisons.

For the MNIST image completion task, we report the average wall-clock training time per epoch, peak GPU memory usage, and total parameter counts. To ensure a fair comparison, all models were trained with a fixed batch size of 128. The results, presented in Table 13, demonstrate WARP's notable efficiency. Despite having a comparable number of parameters to the Transformer model, WARP requires significantly less GPU memory—on par with the much simpler GRU and LSTM architectures—and achieves the fastest training time.

Table 13: Training efficiency comparison on the MNIST image completion task. We report the average wall-clock time per epoch, peak GPU usage, and the number of learnable parameters. WARP is the most efficient in terms of both time and memory.

| **Model** | Avg. training time per epoch (seconds) | Peak GPU usage (GB) | Parameters (M) |
|-----------|----------------------------------------|---------------------|----------------|
| GRU | 57.04 | 4.49 | 1.69 |
| LSTM | 59.22 | 4.95 | 1.70 |
| S4 | 61.53 | 12.60 | 1.71 |
| Transformer | 18.62 | 10.03 | 1.69 |
| **WARP** | **45.22** | **2.89** | **1.69** |

A similar analysis was conducted for the UEA benchmark datasets, with results detailed in Table 14. For these experiments, the batch size was fixed to 32 across all models. The table provides a detailed breakdown of the training time, memory usage, and model complexity for WARP on each dataset.

Table 14: Detailed training metrics for WARP on the UEA benchmark datasets.

| Dataset | Training time per epoch (s) | Peak GPU usage (MiB) | Num. of epochs | Training batches per epoch | Parameters (M) |
|---|---|---|---|---|---|
| Worms | 10.29 | 14598 | 1000 | 6 | 5.697 |
| SCP1 | 0.92 | 654 | 5000 | 13 | 0.476 |
| SCP2 | 3.85 | 2866 | 5000 | 9 | 17.34 |
| Ethanol | 2.22 | 1536 | 6500 | 12 | 4.681 |
| Heartbeat | 6.50 | 4354 | 1500 | 9 | 75.02 |
| Motor | 2.46 | 2558 | 2000 | 9 | 4.503 |

It is important to note that our implementation of WARP, along with the GRU, LSTM, and S4 baselines, utilises JAX. In contrast, the Transformer and all other baselines is implemented in PyTorch. This difference in framework can influence performance measurements. Following standard practice, we exclude any one-time JIT-compilation costs from the reported wall-clock times.

### E.4 NORMALISED TIME CORRELATION ON DYNAMICS RECONSTRUCTION

Let's analyse when the root network takes as input exclusively the normalised time. In that case, WARP uses a diagonal readout matrix $\theta_t(\tau)$ as seen in Fig. 9(a) to self-decode the hidden states. This implies that post training, the weights $\theta_t$ and the time $\tau = t/(T-1)$ should be correlated. We confirm this hypothesis by plotting the correlation coefficient between the vector $\theta_{0:T}$ and all time points across all samples in the test set. We observe a strong either positive or negative correlation between the two quantities (see Fig. 9(b)).

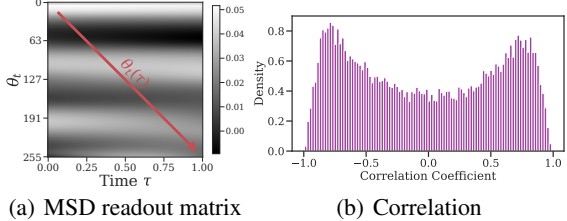

(a) MSD readout matrix      (b) Correlation

Figure 9: (a) Example "readout" matrix on the MSD problem for all time steps $t$ at all times $\tau$, highlighting WARP's diagonal decoding direction $\theta_t(\tau)$; (b) Correlation between the root network's weights $\theta_t$ and the time $\tau$ on the MSD problem; indicating strong linear dependence between the two.

### E.5 SPECTRAL ANALYSIS

To understand the dynamic properties and memory mechanisms learned by our model on the MSD task, we conduct a spectral analysis of its state transition matrix $A$. This analysis is crucial for visualizing how the network learns to retain information over long time horizons. We are looking for eigenvalues clustered near the unit circle ($|\lambda| = 1$), as this indicates a capacity for long-term memory without vanishing or exploding gradients. The analysis in Fig. 10 reveals the model successfully learns to preserve long-term dependencies by maintaining the vast majority of its eigenvalues directly on the unit circle. The minor spill-over ($|\lambda| > 1$) is effectively managed by gradient clipping during training (see Appendix B.3).

### E.6 ROBUSTNESS TO NOISE

We evaluate robustness on the MSD dataset by corrupting the input trajectories with increasing levels of Gaussian noise $\eta$, rescaled such that $\eta = 1$ corresponds to a standard deviation of 39 (the maximum absolute value encountered in the dataset). We employ four metrics to compare the generated trajectories to their noisy ground truths: MSE, RMSE, MAE, and MAPE, as defined in Appendix C.3. The results in Fig. 11 reveal that WARP exhibits robust performance from minuscule ($\eta < 10^{-1}$) up to moderate noise levels ($10^{-1} < \eta < 10^1$), after which all error metrics increase

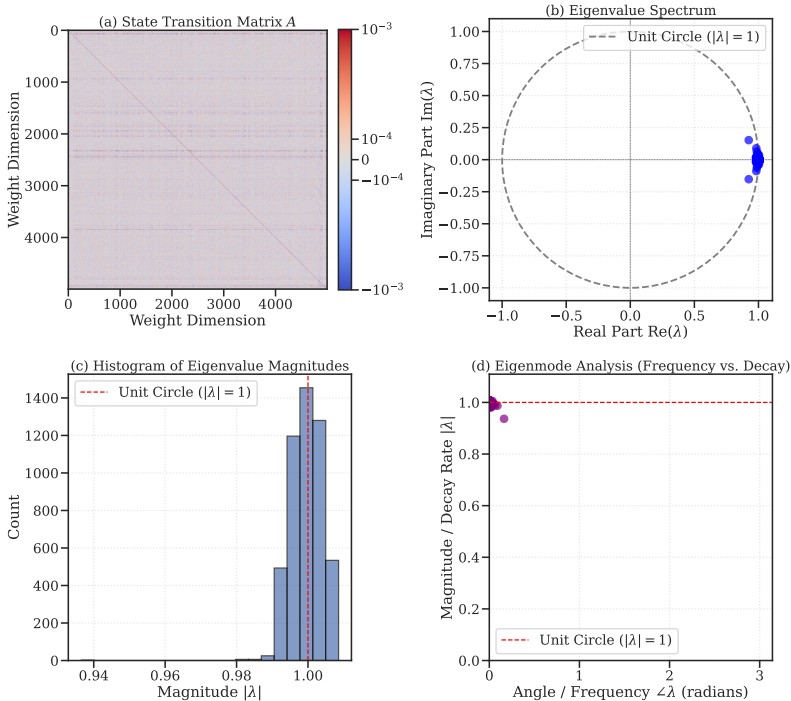

Figure 10: Spectral analysis of the learned state transition matrix $A$ on the MSD task. In both cases, the model successfully learns to place eigenvalues on the unit circle to achieve long-term memory, as observed in subplots (b), (c) and (d). For the visualization in (a), the colormap is intentionally saturated at a low absolute value of $10^{-3}$. This is necessary because the diagonal elements ($\approx 1.0$) are several orders of magnitude larger than the learned off-diagonal couplings.

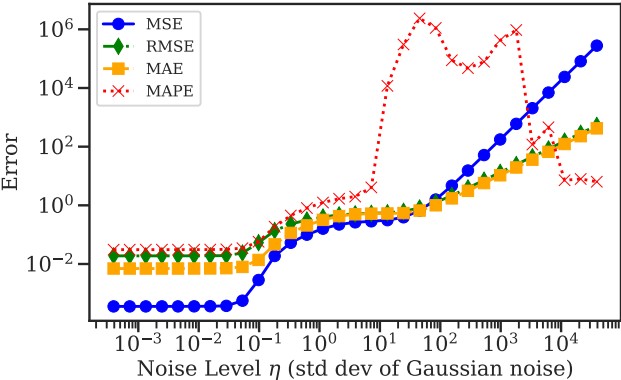

Figure 11: Evaluating performance in noisy scenarios. These results illustrate WARP's robustness to increasing levels of Gaussian noise on the MSD dataset, as measured by several metrics.

sharply by several orders of magnitude, indicating a critical threshold beyond which the model fails catastrophically.

### E.7 ABLATION STUDIES

We briefly discuss several experiments carried out to gain insights into our model. For all ablation studies, experimental protocols like training hyperparameters are presented in **Appendix D**. Figures and Tables in this section are captioned with the corresponding paragraph title.

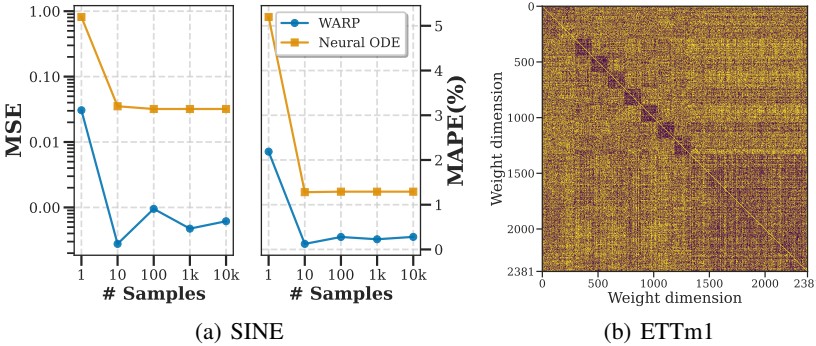

(a) SINE          (b) ETTm1

Figure 13: **(a)** Sample efficiency on SINE. **(b)** A dense weights-to-weights $A$ matrix on ETTm1.

**Eliminating the root network.** The root network $\theta_t$ is integral to the efficacy of WARP. Although not an absolute prerequisite for the WARP-Phys variant, it nonetheless persists as a pivotal constituent of our framework. Illustratively, the omission of $\theta_t$ in favour of directly fitting $\varphi$ for the SINE modelling problem results in a catastrophic degradation in model expressivity.

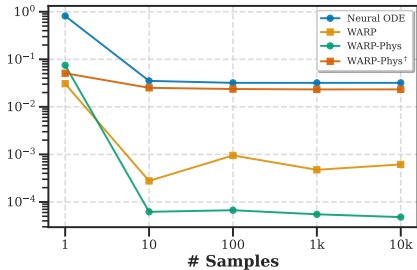

Figure 12: Eliminating the root network — Test-set MSEs on the SINE problem. The omission of $\theta_t$ in favor of directly fitting $\varphi$ (which we call WARP-Phys$^\dagger$) results in a catastrophic degradation in model expressivity. Performance is almost as bad as the Neural ODE analysed in Fig. 13(a).

**Initial network configuration.** Since WARP's weight trajectory is driven by the changes in the signal and not the signal itself (see Eq. (1)), it is important to have an expressive initial hypernetwork $\phi : \mathbf{x}_0 \mapsto \theta_0$, which embeds the initial tokens into suitable weight spaces [52]. Our empirical investigations reveal that sidestepping this component substantially curtails the model performance on complex synthetic benchmarks, such as MSD-Zero, and on real-world datasets, including ETT.

Table 15: Initial network configuration — Empirical investigations reveal that sidestepping $\phi$ in favor of directly learning $\theta_0$ curtails the model performance on complex synthetic benchmarks, such as MSD-Zero, and on real-world datasets, including ETTm1.

| PROBLEM | WITH $\phi$ | WITH $\theta_0$ |
|---|---|---|
| MSD-Zero | 0.32 | 1.02 |
| ETTm1 | 0.02 | 1.25 |

**Data efficiency.** With $L = 1$, the SINE benchmark is a challenging initial value problem. At equal (root) neural network parameter counts, we vary the number of training samples, and we plot MSE and MAPE test metrics for WARP and the Neural ODE [18]. The results, depicted in Fig. 13(a), not only show improved performance across data regimes, but they also indicate that more data is not necessarily better for WARP's performance, suggesting potential for monotone learning [13].

**Dense state transitions & Channel mixing.** The total parameter count of our model is quadratic in the root network's dimensionality $D_\theta$. Specifically, attempts to replace $A \in \mathbb{R}^{D_\theta \times D_\theta}$ with diagonal or low-rank approximations have resulted in remarkably less expressive models, thus solidifying its dense nature, as illustrated in Fig. 13(b), as a key component of our framework.

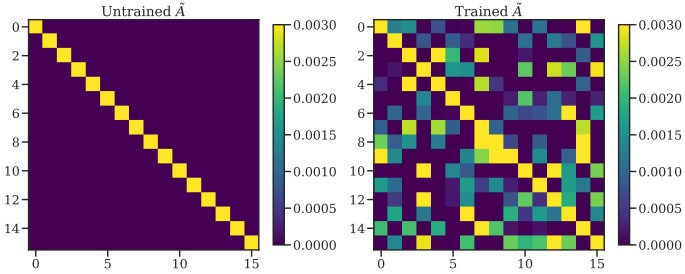

Figure 14: Dense state transitions & Channel mixing — Attempts to replace $A \in \mathbb{R}^{D_\theta \times D_\theta}$ with either a diagonal or a low-rank approximation $\tilde{A}$ result in less expressive models. We observe here a low-rank $\tilde{A} \in \mathbb{R}^{16 \times 16}$ on the ETTm1 problem, such that $A = P\tilde{A}Q$, with all quantities in the right-hand side learnable.

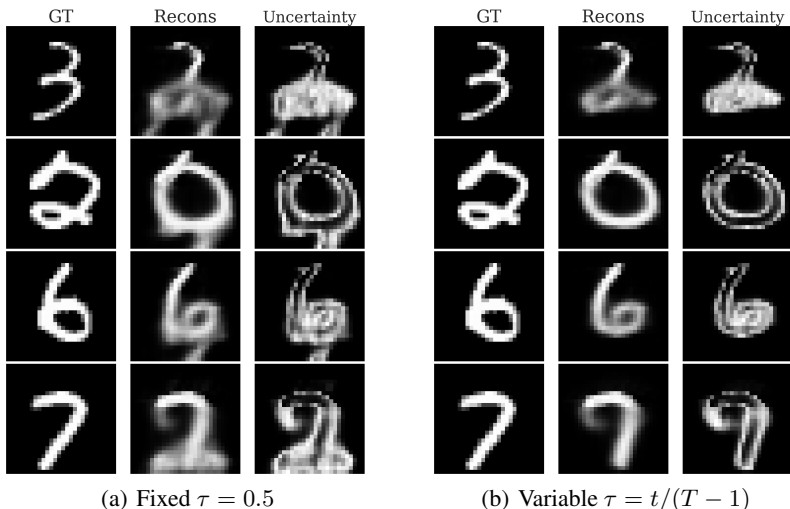

(a) Fixed $\tau = 0.5$          (b) Variable $\tau = t/(T-1)$

Figure 15: Root network evaluation — When using normalised time as the coordinate system, if we fix the evaluation point $\tau$, we observe mild degradation in the qualitative results. While these figures are shown for MNIST with $L = 300$, the behaviour is observed across problems, including dynamical systems like MSD (see Fig. 9(a)). **GT** stands for the Ground Truth, **Recons** is for the Reconstruction/Completion, and **Uncertainty** is the model-outputted standard deviation.

Table 16: Positional Encodings (PE) ablation — We report the classification accuracy (%) of WARP with and without PE on the UEA datasets. The results show a consistent performance drop when PE is removed, underscoring its importance for long-range dependencies such as Worms and Motor.

|  | Worms | SCP1 | SCP2 | Ethanol | Heartbeat | Motor |
|---|---|---|---|---|---|---|
| with PE | $70.93 \pm 2.7$ | $83.53 \pm 2.0$ | $57.89 \pm 1.4$ | $32.91 \pm 4.2$ | $88.65 \pm 1.9$ | $56.14 \pm 5.1$ |
| w/o PE | $60.98 \pm 3.1$ | $80.00 \pm 2.0$ | $57.89 \pm 1.5$ | $31.65 \pm 0.8$ | $77.42 \pm 2.2$ | $50.88 \pm 2.3$ |

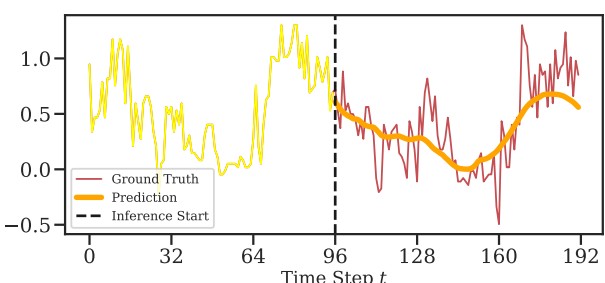

Figure 16: Ablation of the reparametrisation trick — On the electricity problems, if stochastic sampling during training is not used, the model only predicts the mean of the distribution, thereby ignoring high-frequency components or noise in the signal. Here, this is illustrated with a prediction on the ETTm1 test split.

# F  VISUALISATIONS

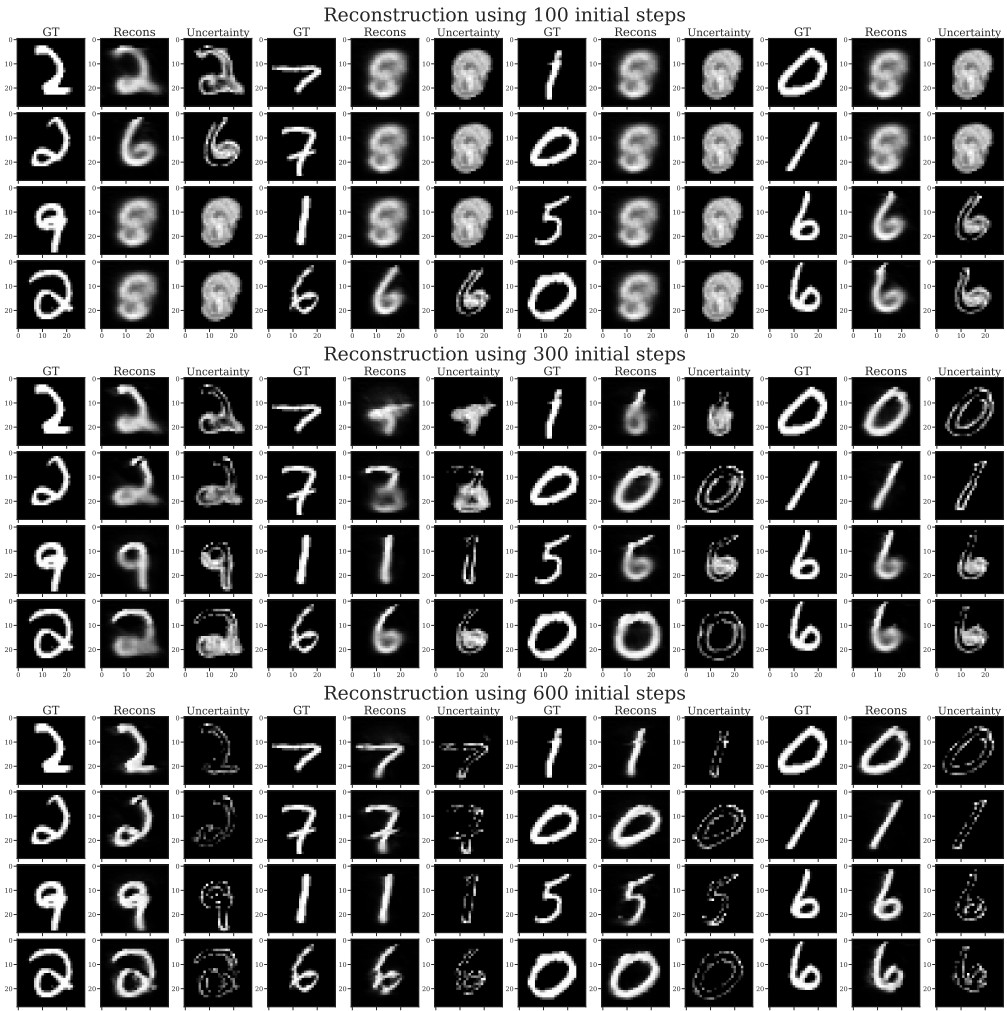

Figure 17: Completed images from the MNIST test set using WARP. The same set of images is shown across three settings: **(Top)** $L = 100$, **(Middle)** $L = 300$, **(Bottom)** $L = 600$. Along the columns, we show 4 groups of results, each with Ground Truth (**GT**), Reconstruction (**Recons**), and **Uncertainty**, resulting in 12 total columns. As our model sees more steps, its forecasting improves and its uncertainty decreases.

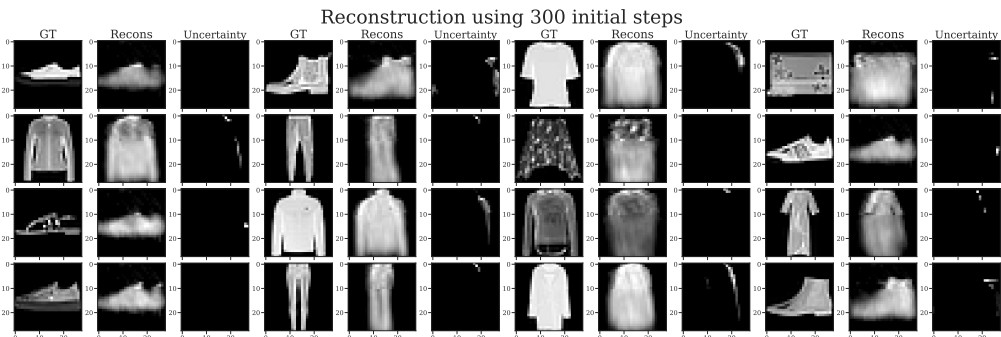

Figure 18: Completed images from the Fashion MNIST test set using WARP.

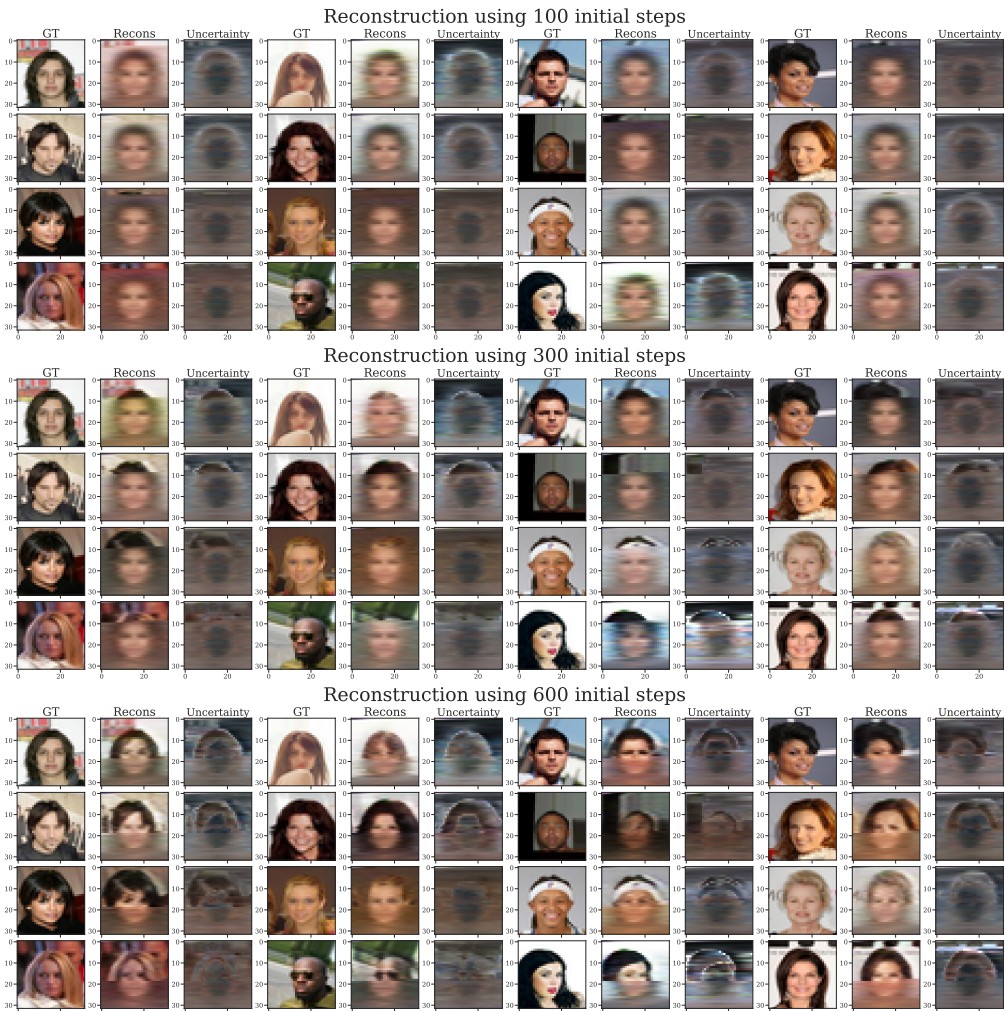

Figure 19: Completed images from the CelebA test set using WARP at various context lengths.

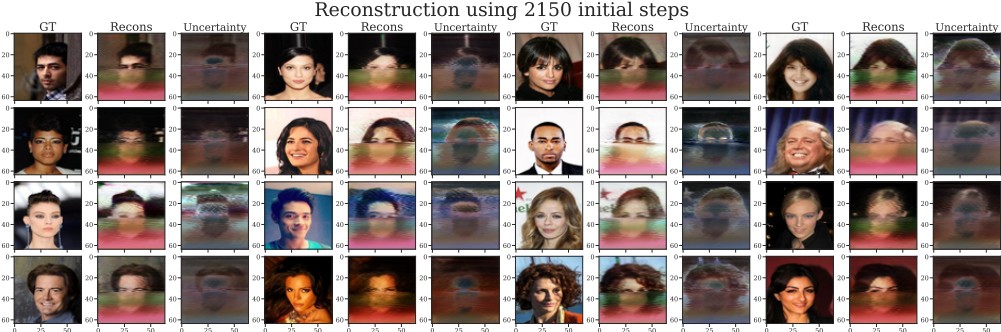

Figure 20: Completed images on the CelebA test set at high-resolution ($T = 64 \times 64 = 4096$), using positional encoding [92]. This illustrates WARP's suitability for long-range dependencies.

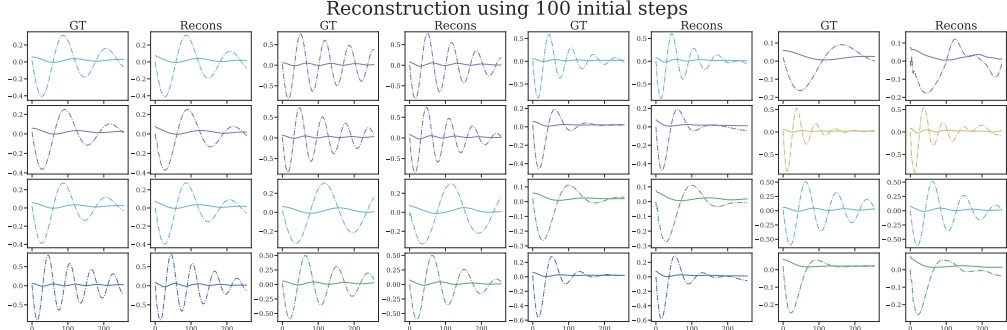

Figure 21: Completed sequences from the MSD test set using WARP.

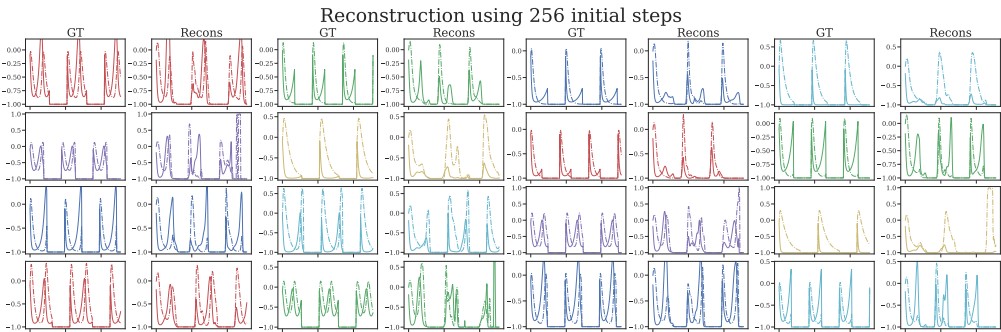

Figure 22: Completed sequences from the LV test set using WARP.

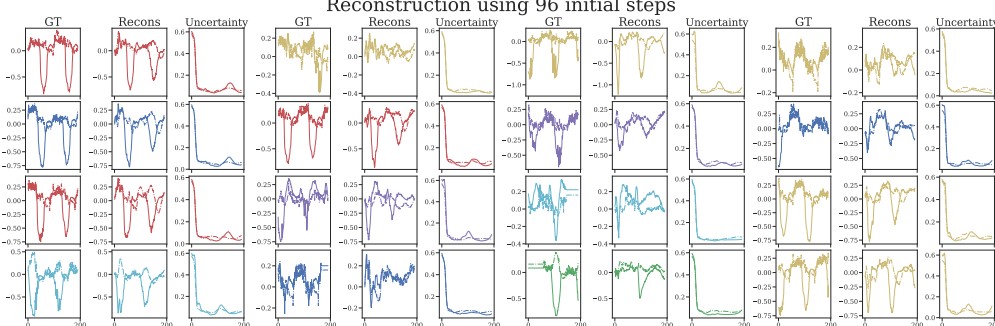

Figure 23: Completed time series from the ETTm1 test set using WARP.

