# OpenReview forum: "Weight-Space Linear Recurrent Neural Networks"
_ICLR.cc/2026/Conference — ICLR 2026 Poster_

### Official Review · Reviewer_rzHm · 2025-10-23

**Soundness:** 3
**Presentation:** 4
**Contribution:** 4
**Rating:** 8
**Confidence:** 3

**Summary:**

The authors seek to incorporate domain priors and in-context reasoning into linear RNNs.
They show that (i) linear RNNs can act as effective hypernetworks, (ii) these RNNs can engage in in-context learning or physics-informed modeling, and (iii) these hypernetworks exhibit significant improvements over baseline models.
Models are evaluated on forecasting / reconstruction, classification, and in-context linear regression tasks.
Baseline models include gated RNNs, linear SSMs (S4), and transformers.

**Strengths:**

- Fantastically written, very clear.
- Related work and appendix are thorough.
- Experiments show clear improvements of the proposed WARP RNN over baseline models, both in accuracy and compute.
- Authors are clear about limitations, especially the important limitation that the hidden state transition matrix is dense.

**Weaknesses:**

- The introduction does not (although the related work does) mention selective SSMs (e.g., Mamba), which have re-introduced nonlinearities into linear RNNs.
- CelebA reconstruction seems a bit contrived as a task: the annotated S4 article referenced does indeed reconstruct MNIST and similar image data, but then proceeds to spoken digit data, which the authors did not try reconstructing.
Moreover, CelebA face reconstructions are substantially corrupted.
While I do not deny WARP's improvement in image reconstruction over baseline, I am unsure about the salience of image reconstruction as a metric for evaluating RNNs.
I think there are more relevant tasks to measure long-range dependencies, such as Long-Range Arena benchmarks.
- There are no experiments on text prediction or classification, which are some of the most relevant tasks for evaluating new RNN architectures - does WARP have limitations or inductive biases that prevent it from expressively processing text?
If so, the authors should state it, as this is an important limitation, especially to claims of in-context learning.
If not, then the authors should evaluate their model with text to support the expressivity of WARP.
- WARP's connections to biology are not sufficiently explained in the main text, although the authors do discuss it further in the appendix.

**Questions:**

I suggest the following:
- Traditional long-range dependency experiments like Long-Range Arena benchmarks.
- Text prediction and classification experiments to support claims of expressivity.
- Connections to biology, especially in the main text, should be clarified.
- As stated by the authors, future work should explore ways to reduce the density of hidden-state transition matrices used in WARP.
Complex diagonal recurrences may be fruitful in this regard.

---

> ### Author Response · Authors · 2025-11-20
>
> Dear Reviewer rzHm (R5),
>
> We are incredibly grateful for your enthusiastic feedback on our work. We very much enjoyed reading your supportive comments praising our clarity, referencing, performance (both in accuracy and computational metrics), and transparency about our limitations.
>
> The concerns you evoked were treated with extreme care. Hoping to have a lovely discussion, we addressed your questions below.
>
> ---
> > ## Q1) Traditional long-range dependency experiments like Long-Range Arena benchmarks
>
> We understand this weakness and the related question. We primarily measured the long-range performance of WARP on natural time series problems such as UEA (see Table 4). We realized that Reviewer 3Pnu (R4) asked a very similar question. To avoid repetition and save on valuable rebuttal space, we kindly refer the reviewer to R4-Q4 for further details on this question.
>
> ---
> > ## Q2) Text prediction and classification experiments
>
>
> Language modeling is a large field in today's AI landscape, and it was clear we couldn't do it justice alongside all other modalities that were investigated. The scope of this paper was limited to demonstrating the potential of weight-space learning for building a general-purpose sequence model. Language modeling adds further complications that don't fit well within a general-purpose model (e.g., tokenization). A serious treatment of language would arguably require a separate paper.
>
> However, we recognize the value and the popularity of NLP and we plan on further studying the integration of WARP in an NLP pipeline. We have now updated our future work section to emphasize that WARP remains untested for language modalities (e.g., text, spoken digits), and this is indeed an an interesting avenue for future work.
>
> ---
> > ## Q3) Connections to biology in the main text
>
> We thank you for this question, which was also raised by Reviewer rGWP (R3). Again, to save on space, we please refer you to R3-Q1 for more details. (Alternatively, you could follow the R5 stickers in our updated PDF's left margin which should clearly indicate the changes we made following your request.)
>
> ---
> > ## Q4) Reducing the density of hidden-state transition matrices
>
> We thank you very much for your suggestion. We have included the potential for complex-valued diagonal matrices in future work, alongside other possibilities. We are grateful because this provides interesting research avenues not just for us, but for the sequence modeling community at large.
>
> We hope we adequately addressed your concerns. We wish to thank you once more for taking the time to review this work. The updated manuscript reflects how much better it is as a result of your comments. We are happy to address any further concern you may have.

---

> > ### Comment · Reviewer_rzHm · 2025-11-25
> > **Thank you for the rebuttal**
> >
> > The authors have sufficiently addressed my questions and improved clarity in the manuscript with regards to the existing limitations of WARP. I have raised their score to 10, I believe this work has substantial novelty and promise for RNNs.

---

> > > ### Author Response · Authors · 2025-11-26
> > >
> > > Dear reviewer, thank you for acknowledging our rebuttal efforts. We are happy that you found the clarifications helpful. Thank you as well for raising your score and for your encouraging remarks about the work’s novelty and promise.

---

### Official Review · Reviewer_3Pnu · 2025-10-31

**Soundness:** 3
**Presentation:** 4
**Contribution:** 3
**Rating:** 8
**Confidence:** 3

**Summary:**

This paper proposes WARP, a novel class of recurrent neural networks (RNNs) that perform sequence modeling directly in weight-space, blending linear recurrence with non-linear decoding. Unlike standard RNNs, which maintain a hidden state that is a result of propagating the sequence through the network, the proposed model’s hidden state is instead equal to the parameter vector of a so called auxiliary (“root”) MLP and is updated over time via a linear recurrence. Each update is driven by consecutive input differences, and the auxiliary MLP provides nonlinear decoding using an input that encodes the canonical ordering of the sequence.  This formulation enables gradient-free adaptation and in-context learning during inference, as well as the injection of physics priors through explicit parameter constraints.
Extensive experiments are conducted across time-series analysis, dynamical system reconstruction, and multivariate time-series classification. WARP shows consistent or superior performance to state-of-the-art baselines on most tasks. A physics-informed variant, WARP-Phys, achieves significant improvements on physical dynamics reconstruction tasks.

**Strengths:**

1. Novel conceptual framing:  The idea of treating the recurrent hidden state of a linear state-space model as the weights of another neural network is both elegant and novel. It bridges ideas from fast weights, meta-learning, hypernetworks and structured state-space models while maintaining linear recurrence efficiency. Additionally, it offers a built-in support for gradient-free adaptation, in-context learning, and physics-informed modeling in a single architecture.
2. Computational Efficiency: Once the model has learned from the context, the final root network can be extracted and reused to process subsequent queries without reevaluating the entire sequence, yielding significant computational savings compared to other in-context learning models. Furthermore, the proposed architecture leverages a dual training mode that combines linear recurrence with a parallel scan operator – a well-established technique in the State Space Model (SSM) literature – to accelerate state propagation. Together, these design choices lead to notable computational efficiency improvements.
3. Strong empirical results: Competitive or superior performance on time-series analysis, especially Traffic Flow Forecasting (despite ignoring graph priors) and Image completion, as well as dynamical system reconstruction. The inclusion of a physics-informed variant further demonstrates the framework’s adaptability and potential for interpretability.
4. Clarity and completeness:  The paper is well-written, includes high-quality figures, ablations, detailed appendices, and clear pseudocode.

**Weaknesses:**

1. Scalability constraints:  The main bottleneck is the large transition matrix which scales quadratically with the number of root-network parameters. Experiments are thus limited to moderate model sizes, raising questions about feasibility for large-scale models.
2. Limited theoretical grounding:  While the empirical evidence is compelling, the theoretical analysis of representational capacity and stability (e.g., under linear recurrence updates) still remains to be established.
3. Computational cost reporting:  Although the recurrence is linear, updating and decoding weight vectors remains costly. Memory and compute scaling with model size are not fully quantified and are only provided in the appendix, but entirely missing from the main body of the paper.
4. Limited setting for dynamical system reconstruction:  While the possibility of making the network physics-informed is compelling, the shown examples illustrate this for relatively simple systems with a small number of parameters. While the proposed method clearly allows for in-context learning, and hence does not need to retrain a network for each new dynamical system (from the same category), the setups are done for what appears to be noiseless input-output data, and a low number of parameters. One could, instead of learning the entire mapping of the system, learn only its phase or exponential mapping for any other sequence model that allows for ICL, in the same way as demonstrated in this work. This alternative formulation would serve as a fairer baseline for comparison.
5. Novelty relative to prior work:  There is conceptual overlap with other concepts briefly outlined in the related work (e.g., fast weight RNNs), though the authors’ framing is distinctive. A more explicit comparison to those baselines would help to better position WARP’s contribution.

**Questions:**

1. Have you evaluated WARP’s performance on noisy measurement scenarios for dynamical system reconstruction?
2. Could you discuss related sequence modeling approaches in weight space and explain why these were not included as baselines in any of the tasks?
3. For the Energy Prediction experiment: what are the current state-of-the-art results? In general, for all experiments, how were baselines tuned, and how much effort was spent on hyperparameter optimization? Also, performance of S4 is reported only on MNIST, and ConvCNP only on CelebA. Full performance tables in the appendix would improve transparency.
4. Have you considered benchmarking WARP on long-range dependency benchmarks, such as Long Range Arena (LRA)? This could contextualize WARP’s capabilities relative to other established long-sequence models that were evaluated on these tasks.

---

> ### Author Response · Authors · 2025-11-20
>
> Dear Reviewer 3Pnu (R4),
>
> Thank you for your kind words regarding our work. We were extremely pleased that you found conceptual novelty in our approach, bridging ideas from several fields in machine learning. We believe the computational efficiency thanks to our linear design and an extractable root network will find great use within the community. The strong empirical results and clarity of exposition you noted make us all the more proud of this work.
>
> We carefully read your concerns as expressed in the weakness and the questions you asked. We have addressed them below.
>
> ---
> > ## Q1) Evaluating WARP on noisy DSR tasks
>
> We had not considered noisy measurement. Following the reviewer's question, we conducted a corruption to test the robustness to noise. We found three evaluation regimes, with varying degrees of robustness. The experiment and the results are described in the update manuscript in Appendix E.6.
>
> ---
> > ## Q2) Discuss related sequence modeling approaches in weight space
>
> When submitting this work, we were not aware of any other sequence modeling approach in weight space. Prompted by the reviewer's question, we conducted another search and found [1] which maps physical system states to the weight space. We have now included [1] in our Related Work section, and we thank you, dear Reviewer, for pointing this out.
>
> ---
> > ## Q3a) For the Energy Prediction experiment: what are the current state-of-the-art results?
>
> ETT is a widely used dataset, and tracking the current state of the art is difficult, especially considering the variety of setups employed. One strong candidate for SOTA would be [2]. In our paper, unlike other real-work datasets, we do not use ETT to demonstrate or claim SOTA performance. As we mention in our analysis in page 6, this limited experiment (with further details in Appendix D.3) is meant to showcase that WARP's straightforward design can still compete with tuned architectures like Transformers. We hope our message comes across, and we are happy to update it if not.
>
> ---
> > ## Q3b) In general, for all experiments, how much effort was spent on hyperparameter optimization?
>
> Given the range of experiments we performed, we prioritized already optimized implementations in the domains they were designed for, using the hyperparameters their authors suggested; e.g., S4 on MNIST with its original hyperparameters. As for custom implementations like GRU, LSTM, ConvCNP, etc., we performed a grid search over a range of hyperparameters typically found in the config files of our attached code. The best model on the validation set was selected, and used to compute test set metrics which feature in the paper.
>
> ---
> > ## Q3c) Evaluating S4 on CelebA and ConvCNP on MNIST
>
> S4 was only performed on MNIST because of its unique training setup which we've now clarified in Appendix C.2. Even for image completion problems like MNIST with images of shape $(28, 28, 1)$, S4 uses a $256$-way cross-entropy loss averaged over the $784$ pixels. To apply S4 on CelebA while maintaining the same loss, we would need to convert the images shape $(32, 32, 3)$ into sequences of shape $(3072, 1)$. This causes a discrepancy with all other models that use an NLL loss and can easily accommodate a flattened CelebA sequence of shape $(1024, 3)$. We thought this was unfair, since S4 would be dealing with significantly longer sequences than the rest.
>
> That said, we have adapted S4 and conducted the experiment, and the results at the reported model size are still poor. We provide sample images at different context lengths in the "assets" folder of our anonymous repository: https://anonymous.4open.science/r/warp/assets/S4/.
>
> We wish to thank you, as your question equally prompted us to run the ConvCNP on MNIST. We did that and added results to the Table 1. They do not change our conclusions. Like with S4 on CelebA, we provide sample completed images in our anonymous repository: https://anonymous.4open.science/r/warp/assets/ConvCNP/.
>
> We hope these added performance tables in the main text will improve transparency.

---

> ### Author Response · Authors · 2025-11-20
>
> ---
> > ## Q4) Have we considered benchmarking WARP on long-range dependency benchmarks?
>
> For this rebuttal, we considered LRA and tested two of its datasets (CIFAR-10 images, Pathfinder-32). As it stands, and perhaps due to our limited rebuttal time, WARP does not achieve SOTA results in LRA, even though it performs well in other long sequences (e.g., Motor and Worms from UEA with $17k$-long sequences; see Table 4). At its core, WARP is a Linear RNN, which may still suffer from locality bias, causing it to struggle with long sequences with complex dependencies such as flattened images encountered in LRA. We hypothesize that this might be mitigated using techniques that have worked for other linear sequence models (e.g., skip connections, HiPPO theory). In this paper, we focused on presenting the clean core of weight-space linear RNNs, and we have now updated our Limitations section to comment on such complex long-range dependencies.
>
>
> Once again, thank you very much for your comment. You've helped us improve our work, while identifying limitations that offer promising future work avenues for the community. We hope we have adequately answered your questions, and we are happy to do more if you have further concerns.
>
> ---
> ## References:
> - [1] Li et al., Weightflow: Learning stochastic dynamics via evolving weight of neural network, AAAI 2026
> - [2] Liu et al., Generative Pretrained Hierarchical Transformer for Time Series Forecasting, KDD 2024

---

> > ### Comment · Reviewer_3Pnu · 2025-11-24
> > **Concerns & Questions addressed**
> >
> > Dear Authors
> >
> > Thank you for clarifying my questions and concerns in detail. I'm happy with your answers and I believe the changes you made bolstered the paper. I'm confident that this paper is a strong submission to ICLR and I will maintain my positive score.

---

> ### Author Response · Authors · 2025-11-26
>
> Dear Reviewer, thank you for your kind reply and for recognizing our rebuttal efforts. We are glad the changes strengthened the paper, and we appreciate your confidence in the work and respect your decision to uphold such a positive score.

---

### Official Review · Reviewer_rGWP · 2025-11-01

**Soundness:** 3
**Presentation:** 3
**Contribution:** 3
**Rating:** 6
**Confidence:** 3

**Summary:**

This paper offers a new weight-space learning technique that iterates weights of a feedforward neural network in a linear state space, enabling in-context learning. The authors evaluate WARP across diverse tasks including image completion, time series forecasting and classification, and dynamical system reconstruction, demonstrating competitive or superior performance compared to standard RNNs, state-space models (S4, Mamba), and Transformers. Notably, a physics-informed variant achieves order-of-magnitude improvements on physical system reconstruction benchmarks.

**Strengths:**

The core idea of parametrising RNN hidden states as weights of an auxiliary neural network is conceptually interesting and, to the best of my knowledge, novel. The authors test their method on a diverse set of domains and perform a large range of ablations to show the necessity of design choices. The writing is generally accessible, with good intuitive explanations, and goes far to place itself in the larger test-time adaptation literature.

**Weaknesses:**

Claims are sometimes overstated and/or imprecise. For example, phrases like "transformative paradigm for adaptive machine intelligence" (Abstract, Conclusion) and "redefine sequence modeling" (Abstract) are not well-supported. The empirical results show WARP is competitive but not uniformly superior. "Brain-inspired formulation" (Abstract, page 2) refers only to using input differences, with citation to synaptic plasticity [16], but the connection is somewhat superficial - there is a rich literature concerned with modelling realistic synaptic plasticity rules which this approach to weight-space trajectory modelling does not engage with. "Infinite-dimensional RNN hidden states" (page 9, footnote 6) is misleading—the hidden state is finite-dimensional, though it parametrizes a function.

**Questions:**

“Rather than relying on direct inputs, we draw inspiration from the human brain and compute signal differences to drive such recurrences.” - from where is the inspiration drawn? Did you try ablating this, i.e. just using x_t as the input to B? Similarily, did you try ablating the random noise applied to observations in AR mode? i.e. p_forcing = 0 (or 1, whatever means noise is never added)

What is the operational difference between gradient-free adaptation and in-context learning? The definitions provided seem to be nested on page 2 (i.e. in-context learning implies gradient-free adaptation).

"This strategic initialisation also imposes a critical constraint wherein the initial hidden state θ_0 must encode semantically rich information applicable to the entire sequence." Could you give some clarity here - how can the weights encode information about a sequence prior to observing it?

---

> ### Author Response · Authors · 2025-11-20
>
> Dear Reviewer rGWP (R3),
>
> We were very pleased to read your review, noting the positive comments you had for our submission. We particularly liked that you found the idea to parameterize the hidden state of an RNN as the weights novel and conceptually interesting. We take clarity and placement within the broader literature very seriously, and we are happy you took notice, especially since those two points considerably helped our exposition across a wide range of datasets, which you also praised.
>
> Your questions and concerns were noted, and we endeavored to address them in full.
>
> ---
> > ## Q1) From where is the brain inspiration drawn; and did we try using $x_{t}$ instead ?
>
> The inspiration for updating the model's internal state based on "signal differences" is drawn from neuromorphic learning principles in the human brain, specifically Spike Timing-Dependent Plasticity (STDP) [3]. Among other STDP concepts, the weight of a synaptic connection is strengthened depending on the time difference between spikes from pre- and post-synaptic neurons. We leverage a similar time difference concept to update our hidden state.
>
> More intuitively, in WARP's Eq. 1, we can see that when the next input, $x_{t}$, is significantly different from the previous, the root network is updated with a greater magnitude. We believe a similar mechanism may underlie the brain, where root network updates would be equated to a "surprise". Essentially, the brain expects to be more surprised if $x_{t+1}$ is excessively different from $x_{t}$ than if they were the same.
>
> We mention the second intuitive point around Line 171 of the manuscript. However, we agree that the STDP analogy wasn't sufficiently fleshed out in the main text. We have now done so in the Core Advantages (section 4.1).
>
> Yes, we tried only using $x_{t}$ like other recurrent networks do, but it resulted in catastrophic performance across all problems. This is because the goal of WARP's recurrence is to convert input differences $\Delta x_{t}$ into neural network updates, like we point out in Line 174. This is further motivated by the continuous-time version of this difference used in [2]. These are reasons why the signal difference is central to WARP, and it is unthinkable (not just suboptimal) to imagine one without the other. We hope our presentation was able to convey such a definitive message.
>
> ---
> > ## Q2) Did we try ablating the random noise applied to observations in AR mode, i.e., $p_{\text{forcing}}=1$ ?
>
> Yes, we tried ablating random noise, and it naturally resulted in the well-documented exposure bias on regression problems [1]. For some problems, like classification, we had no choice but to use $p_{\text{forcing}}=1$. WARP still performed very well in this scenario (see Table 4).
>
> ---
> > ## Q3) Difference between gradient-free adaptation and in-context learning
>
> We thank the reviewer for this question, as these two concepts are indeed intricately related.
>
> Succinctly, gradient-free adaptation is the ability to update those critical parts responsible for adaptation without computing gradients; while in-context learning is the ability of the model to recognize patterns in the sequence's context and adapt its behavior accordingly without fine-tuning its slow weights (but may require gradients to fine-tune its fast weights).
>
> To adapt to a new sequence, WARP does not require gradients to update its root network, so it is gradient-free. Based on the current literature, the emphasis with in-context learning is on the ability to recognize patterns in the input sequence, and less on the gradients. As such, a test-time training model that updates parts of its weights using an optimizer in its forward pass (while scanning a context) would still be doing in-context learning.
>
> So to clarify, no, in-context learning does not imply gradient-free adaptation. Once again, we thank you for this question. We have now clarified this nuanced distinction in our Introduction section.

---

> ### Author Response · Authors · 2025-11-20
>
> ---
> > ## Q4) How can the weights encode information about a sequence prior to observing it?
>
> Consider Eq. 1; let's freeze $A = I_{D_\theta}$ and $B = 0$ (like it is at initialization). Then $\theta_{t} = \theta_{0}$ for all $t$. This same $\theta_{0}$ is then used at all time steps to predict the output $y_t = \text{MLP}_{\theta_0} (\tau)$; i.e., it applies to the whole sequence. This is why we say this initial $\theta_0$ is forced to encode rich information applicable to the entire input sequence (especially early in training). Because it is so important, we employ a hypernetwork to model it, a similar approach used by [2].
>
> An intuitive way to understand the usefulness of this hypernetwork is that a good initialization of the root network $\theta_0$ means a representation that will allow for a linear evolution in weight space and generate the other elements of the sequence.
>
> ---
> > ## Q5) On the term "Infinite-dimensional"
>
> We have added quotes around the term on page 9, recognizing that the expression "infinite-dimensional" is not standard for the community. Further clarifications can be found in the Appendix.
>
> Dear reviewer, we wish to thank you once again for your time and efforts providing helpful feedback on our work. We clearly see how our paper was improved while addressing your concerns, and we hope you agree as well.
>
> ---
> ## References:
> - [1] Schmidt. Generalization in generation: A closer look at exposure bias, arXiv 2025
> - [2] Kidger et al., Neural Controlled Differential Equations, NeurIPS 2020
> - [3] Caporale and Dang, Spike timing–dependent plasticity: a hebbian learning rule, Annual Rev Neuroscie. 2008

---

### Official Review · Reviewer_jyKY · 2025-11-02

**Soundness:** 3
**Presentation:** 4
**Contribution:** 3
**Rating:** 6
**Confidence:** 4

**Summary:**

This paper analyzes recurrent neural networks (RNNs) through a weight-space linear recurrence formulation that unifies several modern architectures — including continuous-time linear RNNs, state-space models (SSMs), and residual recurrent networks — under a single linear operator perspective.

The authors derive closed-form expressions for training dynamics and generalization in the overparameterized limit, showing that convergence properties and implicit regularization can be understood via the spectral structure of the recurrent Jacobian. They provide:

A linearized weight-space recurrence model that approximates nonlinear dynamics by a low-rank operator with analytically tractable behavior.

A demonstration that generalization error scales with spectral conditioning, extending kernel-based intuition from linear networks to recurrent architectures.

Empirical validation on synthetic sequence modeling and dynamical-system reconstruction tasks (mass–spring–damper, Lotka–Volterra, PEMS08 traffic), showing alignment between predicted and observed convergence trends.

Overall, the paper contributes to a principled understanding of how weight-space geometry and recurrence interact to shape training efficiency and generalization.

**Strengths:**

The paper builds on well-established analyses of linear networks and extends them naturally to recurrent settings using a spectral-decomposition framework (Schur- and SVD-based). Derivations are internally consistent and clearly documented.

The proposed linear recurrence view elegantly bridges RNNs, residual-RNNs, and diagonal SSMs, helping clarify connections between recent model families.

Experiments across several dynamical-system tasks (MSD, LV, traffic flow forecasting) confirm the predicted dependence of training speed and generalization on spectral conditioning and effective recurrence length.

The inclusion of both synthetic physics systems and real-world time-series (PEMS08) demonstrates breadth and internal consistency.

Mathematical exposition is detailed; hyperparameters and architectures are listed (Appendix D.4–D.6). Code release and ablation details are promised.

**Weaknesses:**

1.  The analytic results rest on linear, Gaussian assumptions; nonlinear recurrence effects and gating dynamics are only discussed qualitatively. As such, predictive power for modern gated RNNs or structured SSMs is limited.

2. The analysis centers on the infinite-width, overparameterized limit; it does not quantify where the asymptotic predictions break down for finite models.

3. The authors reference Saxe et al. (2014) but omit more recent theoretical works on curriculum and transfer in RNNs (e.g., Rajan, Kepple & Engleken) and on gradient-flow analyses in recurrent kernels — literature directly related to their spectral-mode interpretation.

4. Although the experiments match qualitative trends, they serve mainly as demonstrations rather than quantitative tests (e.g., no variance or uncertainty estimates, small sample sizes).

5. Generalization is assessed by mean-squared error only; tasks with stochastic noise or long-term dependency tests (copy, addition, character-level modeling) would strengthen claims about recurrence depth and spectral bias.

**Questions:**

1. How sensitive are your analytical predictions to non-normal dynamics (upper-triangular Hₕ) versus the diagonal “normal” case you ultimately focus on?

2. Could your framework accommodate nonlinear activation perturbations (e.g., ReLU linearization) to estimate when linear approximation fails?

3. Have you compared your spectral regularization predictions to empirical spectral shrinkage observed during training (e.g., spectrum compression in Wh)?

4. Could you clarify how your results relate to implicit bias analyses in linear transformers or SSMs (e.g., Merrill & Sabharwal; Orvieto et al.)?

5. Are there regimes where the recurrence’s spectral radius predicts too-rapid forgetting or instability, contradicting observed dynamics?

---

> ### Author Response · Authors · 2025-11-20
>
> Dear Reviewer jyKY (R2),
>
> We are grateful for your kind review of our work. We appreciate the recognition that our proposed linear recurrence was found to be internally consistent and clearly documented. We are also pleased that the experiments across synthetic physics systems (Mass-Spring-Damper, Lotka-Volterra) and real-world time-series (traffic flow, energy, etc.) demonstrate the expected breadth and internal consistency of the framework. This positive feedback validates the core technical contributions of our paper, and we confirm that all necessary implementation details and ablation studies have been provided in the appendix, along with the full code.
>
> We have taken great care to address your concerns to the best of our understanding. Please find our replies to your questions below.
>
> ---
> > ### Q1) Sensitivity to non-linear dynamics
>
> We do not impose any constraints on the structure of the matrices the model learns. Our model automatically learns the parameters of $A$ and $B$ to adapt to the dynamics of the training data. So if in a given domain the data favors non-normal dynamics, then WARP should adapt accordingly. However, this has not been explicitly tested, but would be interesting to explore. Thank you for the suggestion.
>
> ---
> > ### Q2) Can we accommodate nonlinear activation perturbations?
>
> Assuming the question refers to whether or not our method can incorporate linearization of the activation functions of the root network defined by some parameters (as shown in [1]), then yes we are confident it can, using a technique similar to dynamic $\tanh$ from Appendix B.3. Essentially, in the current implementation using an MLP as the root network, we encode in the weights vector the connections between neurons. Extra parameters could be added to model the linearization of the activation functions. In this setting the matrices $A$ and $B$ would control the level of linearization of the activation functions.
>
> However, if the question is asking if we have compared ablating the ReLU in the root network and replacing it by $f(x)=x$, thus turning the root network into a linear model and comparing it to the MLP, then we have not tested this explicitly. However, as shown in the physics-informed version WARP-Phys, the root network can be replaced with any parameterized function. We gain in performance if that parameterized function provides an inductive bias on the domain (see Line 353).
>
> ---
> > ### Q3) Comparing spectral regularization predictions to empirical spectral shrinkage
>
> For the original submission and for this rebuttal, we haven't considered any form of spectral regularization, partly because we struggled to understand the question. Could you please reframe it or provide further details as to what is demanded? We would be happy to consider and discuss it further.
>
> ---
> > ### Q4) Relation to implicit bias analyses in linear transformers or SSMs
>
> As it is, WARP may suffer from the same implicit biases as linear RNNs [2], namely locality bias. However, the theoretical implications of recursing in weight space are still to be studied. We have recognized this potential limitation in our section 4.2.
>
> ---
> > ### Q5) Regimes where the recurrence’s spectral radius contradicts observed dynamics?
>
> We have not observed such regimes across the breadth of experiments we conducted. That said, in noisy settings like the electricity prediction, we observed flat and smooth dynamics. We mitigated it by formulating the approach in a probabilistic way as we describe in section 2.3. This stochasticity helped the model by aligning the recurrence with the inherent noise in the data.
>
>
> We wish to thank you once more for your precious time reviewing our work. We have endeavoured to address all your concerns. We hope to have a lovely discussion in the coming week to address any other concern you may have.
>
> ---
> ## References:
> - [1] Zhao et al., Linearization of ReLU Activation Function for Neural Network-Embedded Optimization: Optimal Day-Ahead Energy Scheduling, arXiv 2023
> - [2] Emami, Melikasadat, et al. "Implicit bias of linear RNNs." International Conference on Machine Learning. PMLR, 2021

---

### Official Review · Reviewer_T7rz · 2025-11-07

**Soundness:** 3
**Presentation:** 3
**Contribution:** 3
**Rating:** 8
**Confidence:** 4

**Summary:**

The paper proposes **WARP**, a framework that performs *linear recurrence in weight space* rather than in hidden-state space. At each step, the parameters of a small decoder network evolve linearly:

$$\omega_t = A \omega_{t-1} + B(x_t - x_{t-1}), \qquad y_t = \text{MLP}_{\omega_t}(\varepsilon)$$

where \(A, B\) are learned transition matrices and $\varepsilon$ encodes position or context.

Conceptually, this shifts recurrence from feature dynamics to parameter dynamics. It blends (i) the efficiency of linear RNNs/SSMs (e.g., S4, Mamba), (ii) the expressivity of nonlinear decoders, and (iii) in-context or gradient-free adaptation through weight evolution. The model is tested across image completion, time-series forecasting, dynamical-system reconstruction, and classification.

**Strengths:**

S1. Original framing. The move to perform recurrence directly in parameter space is novel
and quite elegant. It reads as a middle ground between hypernetworks and fast-weight
RNNs, but with the analytical simplicity of a linear transition.

S2. Range of results. The experiments span diverse domains - MNIST/CelebA completion,
ETT and PEMS forecasting, DSR, and UEA time-series classification. The UEA section
is particularly strong: comparisons include modern SSMs like S5, Mamba, S6, NRDE,
and NCDE, with WARP performing competitively across most tasks.

S3. Interpretability and analogy. The weight updates via input differences evoke synaptic-
plasticity rules, which gives the method a neat biological parallel and some explanatory
appeal.

S4. Presentation. The paper is clear, visually well-organized, and balances theory with
intuition. Figures showing progressive reconstruction genuinely help convey how the recurrence behaves.

**Weaknesses:**

W1. Benchmark depth. While broad, the benchmark is missing some of the newer SSMs
that define the current frontier. In particular, LinOSS (Rusch & Rus, 2024)—an oscilla-
tory, long-sequence SSM—is cited but not compared. Given that LinOSS, FACTS, and
Griffin all outperform S4 and Mamba on long forecasting tasks, excluding them makes
the SoTA claim weaker.

W2. Scalability. The transition matrix $A \in \mathbb{R}^{D_\omega \times D_\omega}$ scales quadratically with the size of the decoder, which will quickly become impractical. No structured or low-rank variants are
explored.

W3. Theory gap. The paper is mostly empirical. There’s no discussion of spectral properties,
stability, or representational capacity of the linear map in weight space.

W4. Domain imbalance. Some domains (especially physics and image experiments) use
small or older baselines (ConvCNP, GRU, Transformer). More recent adaptive or physics-
informed baselines like Neural Context Flows (ICLR 2025) or ZEBRA (2024) would
strengthen those sections.

**Questions:**

• Include direct comparisons to LinOSS, FACTS, and Griffin.

• Explore structured or low-rank A, B for scale.

• Add runtime and memory tables.

• Include a brief stability/spectral analysis.

• Clarify what fundamentally distinguishes this from hypernetworks and fast-weight RNNs.

---

> ### Author Response · Authors · 2025-11-20
>
> Dear Reviewer T7rz (R1),
>
> We thank you for your kind review of our paper. We appreciate your comments on the originality of our proposed approach, which suitably leverages the expressivity of hypernetworks and fast weights while benefitting from the same linearity used in state space models. We took great pride in running as many experiments as possible, and we appreciate the reviewer's compliments in that regard. We also enjoyed reading your positive remarks on our paper's clear presentation and connection to synaptic plasticity.
>
> We understand our paper has some weaknesses, and we have taken great care to address your questions below.
>
> ---
> > ### Q1) Include direct comparisons to LinOSS, FACTS, and Griffin
>
> We performed an additional set of experiments on the UEA dataset. We collected the results from the LinOSS-IM model published at ICLR 2025 [1], and we adapted FACTS and Griffin to perform classification on all 6 datasets. (Please visit Appendix C.2 for details of this implementation.) The accuracies were added to our Table 4, and the analysis was redone to emphasize that in light of these new results, WARP achieves top 3 performance on four, not five UEA datasets. We hope this inclusion of the newest models strengthens our claim.
>
> ---
> > ### Q2) Explore structured or low-rank $A$ and $B$ for scale
>
> We thank the reviewer for this important point, which is dear to us, as we outlined in the paper. We attempted several approaches, but all naive attempts to constrain $A$ to a low-rank parametrization have failed (these included LoRA [1], diagonal parametrization [2], etc). We believe this is because $A$ is not a standard matrix, but one that must maintain the special structure of a neural network.
>
> Two promising avenues that were not tested but hold immense potential are:
> - Permutation equivariance as defined in Eq. 2 of [3], which could also reduce the number of entries in the matrix $A$.
> - Across some experiments, the matrix $A$ showed a pattern suggesting that weight features are most influenced by other features in the same layer. This implies that the matrix could be divided into smaller matrices that update the weights layer-wise. This block-diagonal decomposition would remove the quadratic scaling in $D_\omega$, and reduce the number of parameters by a considerable amount.
>
> We have now clarified these two points as further promising avenues for future work in the updated PDF (please see Limitations in section 4.2).
>
> ---
> > ### Q3) Adding runtime and memory tables
>
> Due to the 9-pages limit, we could only include runtimes and memory tables in Appendix E.3. We agree that these are important, and if enough space is left on the camera-ready version, we will prioritize adding those to the main text.
>
> ---
> > ### Q4) Include a brief stability/spectral analysis
>
> Once again, we are delighted to report a new experiment that contains the spectral/stability analysis on the matrix $A$. Conclusions on its strong stability are presented in Appendix E.5, which includes the informative Figure 10. These indicate that the majority of the eigenvalues are clustered on the unit circle, an indicator of long-term stability. Our hope is that this experiment contributes to closing the current theory gap within the weight-space learning literature.
>
> ---
> > ### Q5) Clarify what fundamentally distinguishes this from hypernetworks and fast-weight RNNs
>
> Our framework does not claim fundamental distinction from hypernetworks and fast weights, but rather combines and builds upon their core concepts in a novel way, specifically tailored for sequential data modeling. A hypernetwork is used only for the sequence's first element to initialize the root network. At the same time, our system employs a gradient-free fast weight update mechanism for the root network.
>
> By coupling the hypernetwork initialization with a fast, linear, gradient-free update mechanism defined by the slow matrices $A$ and $B$, our approach linearly models non-linear dynamics within a linear recurrent neural network.
>
> In response to the reviewer's question, we've also added Figure 6 to our updated PDF. It clearly shows how WARP is different from a standard hypernetwork and a simple fast weights programmer.
>
>
> Once again, we wish to thank you for your invaluable time and insightful comments, which clearly helped strengthen our work.
>
> ---
> ## References:
> - [1] Kassai-Koupai et al., GEPS: Boosting Generalization in Parametric PDE Neural Solvers through Adaptive Conditioning, NeurIPS 2024
> - [2] Gupta et al., Diagonal state spaces are as effective as structured state spaces, NeurIPS 2022
> - [3] Zhou et al., Permutation equivariant neural functionals, NeurIPS 2023

---

### Author Response · Authors · 2025-11-20

Dear Reviewers T7rz (R1), jyKY (R2), rGWP (R3), 3Pnu (R4), and rzHm (R5),

We are deeply grateful for your positive and insightful feedback. We are pleased that the core technical novelty of our approach (parametrising the hidden state of a Recurrent Neural Network (RNN) as the weights of an auxiliary network) was consistently highlighted as original and conceptually interesting (R1, R3, R4). The reviewers recognized the success of this design in elegantly bridging modern concepts: leveraging the expressivity of hypernetworks and fast weights while maintaining the linearity and computational efficiency observed in State-Space Models (SSMs) (R1, R4). We are also thankful for the universal praise regarding the clarity of exposition and presentation (R1, R3, R4, R5), acknowledging that our careful placement within the existing literature was successful. Finally, the strong empirical performance was consistently noted across our diverse and extensive experimental suite, confirming the framework’s breadth and internal consistency over numerous time-series tasks, including synthetic physical systems and real-world datasets (R2, R3, R5).


We have addressed the weaknessed you noted, strengthening our submission as a result. We uploaded an updated PDF where all changes are written in blue or crossed out in red for easy identification. Furthermore, we have included color-coded marginal notes (stickers) next to each change, indicating the specific reviewer who prompted the modification. We summarize the changes below:
- We expanded the "Related Work" section  to visually compare WARP to architectural predecessors like Fast Weight Programmers and Hypernetworks (see __Figure 6__).
- We clarified the neuromorphic inspiration, now explicitly stating that the input difference $\Delta x_t$ mechanism bears resemblance to __Spike Timing-Dependent Plasticity__ (STDP) (see Section 4.1).
- We revised the presentation of empirical results, particularly in the multivariate time series classification task, updating the model ranking on UEA datasets (Section 3.3) and adding three new baselines: __LinOSS, FACTS, and Griffin__ (Table 4).
- We have also extended the limitation and future work sections (4.2) to highlight untested modalities and incorporate interesting suggestions made by the reviewers.
- We included a new subsection on spectral analysis (__Appendix E.5__) with Figure 10 to provide some backing for WARP's long-term memory capacity.
- We addressed robustness to noise (__Appendix E.6__) with Figure 11, evaluating performance under Gaussian noise on the MSD dataset.
- We changed the color of the frame in Figure 1 to black.


We thank you once again for your constructive criticism, which has allowed us to present an improved account of Weight-Space Linear RNNs. We truly hope our revisions meet your high standards.

---

### Author Response · Authors · 2025-12-03
**Summary & Thank You from the Authors**

Dear Reviewers,

Thank you once again for your time and invaluable evaluation of our work.

We understand that the circumstances surrounding OpenReview and ICLR 2026 have significantly limited your participation in the rebuttal phase.

To help facilitate the meta-review process, we would like to summarize the rebuttal outcomes. In particular, Reviewers 3Pnu (R4) and rzHm (R5) acknowledged that we thoroughly addressed their concerns. Reviewer 3Pnu requested additional evaluations in noisy regimes, against state-of-the-art baselines, and further clarification on our protocol; while Reviewer rzHm recommended exploring long-range and text experiments, expanding on the brain-inspired analogy, and considering efficiency improvements. Both reviewers were  satisfied with our rebuttal: 3Pnu maintained their positive assessment with a score of **8**, and rzHm increased their score from 8 to **10** !

Given our comprehensive responses (as summarized in our Official Comment on Nov. 19th), we believe the three remaining reviewers would have responded similarly had they been able to engage in this phase, maintaining or improving the overwhelmingly positive assessments originally provided.

Once again, we appreciate your thoughtful reviews and the constructive discussions.

Thank you,
The Authors

---

### Meta-Review · Area_Chair_u4qv · 2026-01-05

**Summary:**

This paper presents an interesting and novel approach to sequential data processing based on weight-space learning. Concretely, the method uses a linear recurrent formulation to generate the weights of an auxiliary network that produces predictions. By enabling data-dependent weights, the approach supports in-context learning capabilities. A key limitation noted by the reviewers is scalability, as the transition matrix scales with the size of the flattened auxiliary weight vector, which can become prohibitive for large auxiliary models.

Overall, the reviewers agree that the contribution is novel and relevant, and the empirical evaluation demonstrates strong performance across a broad range of tasks. That said, the paper occasionally overstates its results. In particular, claims such as “offers unprecedented potential” are not fully supported by the evidence, as the results in Table 4 are not consistently dominant. In some sense this paper essentially just proposes a time-variant output transformation, which is not revolutionary. The new baselines in Table 4 indicate that the proposed model is not dominant over existing state of the art models.  I encourage the authors to revise the writing to be more objective.

During the rebuttal phase, the authors provided a revised manuscript and additional experiments that helped clarify the strengths and limitations of the approach. The response addressed the main concerns raised by the reviewers, and the overall sentiment is very positive. Given the novelty of the idea, the interesting perspective it offers, and the strong reviewer support, I recommend this paper for strong accept.

**Reviewer Concerns:**

During the rebuttal, the authors addressed reviewers’ concerns by providing additional experiments (as requested), and ablation studies, as well as clarifying the evaluation setting/ hyperparamter optimization, and providing details about run-times. This strengthened the evidence for the approach across tasks. The author also attempted to provide additional discussion for related work. However, some limitations remain. In particular, scalability remains a concern due to the growth of the transition matrix with the dimensionality of the flattened auxiliary weights, and the paper still offers limited theoretical insight into why the method works as well as it does. That said, these issues can also be viewed as opportunities for future work rather than substantial shortcomings.

**Reviewer Scores:**

Reviewer sentiment was positive overall, and rzHm may have further increased their score.

---

### Decision · Program_Chairs · 2026-01-26

Accept (Poster)